# Length-independent telomere damage drives post-mitotic cardiomyocyte senescence

Rhys Anderson[1,2,†], Anthony Lagnado[1,2,†], Damien Maggiorani[3,†] ID, Anna Walaszczyk[4], Emily Dookun[2,4], James Chapman[1,2], Jodie Birch[1,2], Hanna Salmonowicz[1,2], Mikolaj Ogrodnik[1,2], Diana Jurk[1,2,5], Carole Proctor[1,2], Clara Correia-Melo[1,2], Stella Victorelli[1,2], Edward Fielder[1,2], Rolando Berlinguer-Palmini[6] ID, Andrew Owens[4], Laura C Greaves[7], Kathy L Kolsky[8], Angelo Parini[3], Victorine Douin-Echinard[3], Nathan K LeBrasseur[8], Helen M Arthur[4], Simon Tual-Chalot[4], Marissa J Schafer[8], Carolyn M Roos[8], Jordan D Miller[8], Neil Robertson[9], Jelena Mann[10], Peter D Adams[9,11], Tamara Tchkonia[8], James L Kirkland[8], Jeanne Mialet-Perez[3,*] ID, Gavin D Richardson[4,**] ID & João F Passos[1,2,5,***] ID

## Abstract

Ageing is the biggest risk factor for cardiovascular disease. Cellular senescence, a process driven in part by telomere shortening, has been implicated in age-related tissue dysfunction. Here, we address the question of how senescence is induced in rarely dividing/post-mitotic cardiomyocytes and investigate whether clearance of senescent cells attenuates age-related cardiac dysfunction. During ageing, human and murine cardiomyocytes acquire a senescent-like phenotype characterised by persistent DNA damage at telomere regions that can be driven by mitochondrial dysfunction and crucially can occur independently of cell division and telomere length. Length-independent telomere damage in cardiomyocytes activates the classical senescence-inducing pathways, p21$^{CIP}$ and p16$^{INK4a}$, and results in a non-canonical senescence-associated secretory phenotype, which is pro-fibrotic and pro-hypertrophic. Pharmacological or genetic clearance of senescent cells in mice alleviates detrimental features of cardiac ageing, including myocardial hypertrophy and fibrosis. Our data describe a mechanism by which senescence can occur and contribute to age-related myocardial dysfunction and in the wider setting to ageing in post-mitotic tissues.

**Keywords** ageing; cardiomyocytes; senescence; senolytics; telomeres
**Subject Categories** Ageing; DNA Replication, Repair & Recombination; Metabolism
The EMBO Journal (2019) 38: e100492

See also: **T Brand** (March 2019)

## Introduction

The role of cellular senescence in tissue maintenance, repair and ageing is currently a rapidly evolving area of research under intense focus. Cellular senescence is classically defined as an irreversible loss of division potential of mitotic cells, often accompanied by a senescence-associated secretory phenotype (SASP). While senescence can be detrimental during the ageing process, it has also been implicated in fundamental biological processes such as tumour suppression, embryonic development and tissue repair (Muñoz-Espín & Serrano, 2014).

Telomere shortening has been proposed as a major inducer of senescence (Bodnar et al, 1998). Telomeres are repetitive sequences

1  Ageing Research Laboratories, Institute for Ageing, Newcastle University, Newcastle upon Tyne, UK
2  Institute for Cell and Molecular Biosciences, Newcastle University, Newcastle upon Tyne, UK
3  INSERM Institute of metabolic and cardiovascular diseases, University of Toulouse, Toulouse, France
4  Cardiovascular Research Centre, Institute for Genetic Medicine, Newcastle University, Newcastle upon Tyne, UK
5  Department of Physiology and Biomedical Engineering, Mayo Clinic, Rochester, MN, USA
6  The Bio-Imaging Unit, Newcastle University, Newcastle upon Tyne, UK
7  Wellcome Trust Centre for Mitochondrial Research, Centre for Ageing and Vitality, Newcastle University, Newcastle upon Tyne, UK
8  Robert and Arlene Kogod Center on Aging, Mayo Clinic, Rochester, MN, USA
9  Institute of Cancer Sciences, CR-UK Beatson Institute, University of Glasgow, Glasgow, UK
10 Institute of Cellular Medicine, Newcastle University, Newcastle upon Tyne, UK
11 Sanford Burnham Prebys Medical Discovery Institute, La Jolla, CA, USA
   *Corresponding author. Tel: +33 5 61325643; E-mail: Jeanne.Perez@inserm.fr
   **Corresponding author. Tel: +44 0 1912418615; E-mail: Gavin.Richardson@ncl.ac.uk
   ***Corresponding author. Tel: +1507 293 9785; E-mail: Passos.Joao@mayo.edu
   †These authors contributed equally to this work

of DNA, associated with a protein complex known as shelterin (de Lange, 2005), which facilitates the formation of a lariat-like structure to shield the exposed end of DNA (Griffith *et al*, 1999), thus protecting the ends of chromosomes from being recognised as DNA damage (d'Adda di Fagagna *et al*, 2003). The current dogma of telomere biology suggests that telomere shortening with each successive cell division eventually disrupts the protective cap, leading to a sustained DNA damage response (DDR) and activation of the senescence programme (Griffith *et al*, 1999). This hypothesis may explain age-related degeneration of tissues maintained by constant contribution of stem cell pools, such as the skin and hematopoietic systems; however, it is insufficient to explain how senescence contributes to ageing in primarily post-mitotic tissues such as the heart. As such, the mechanisms that drive senescence in post-mitotic cells and the contribution of post-mitotic cell senescence (PoMiCS) to tissue degeneration during ageing are an emerging area of interest (Anderson *et al*, 2018; Sapieha & Mallette, 2018).

Clinical heart failure (HF) represents a worldwide problem, and over 50% of all patients exhibit HF with preserved ejection fraction (HFpEF), which is a disease of the older population (Borlaug, 2014). As the population continues to age, the prevalence of HFpEF is increasing, currently at a rate of over 10% per decade (Borlaug & Paulus, 2011). During ageing, the heart undergoes distinct physiological and molecular changes, including cardiomyocyte (CM) hypertrophy and increased fibrosis, which contribute to increased ventricular stiffness causing a reduction in diastolic but not systolic cardiac function (Strait & Lakatta, 2012). Despite the growing need for interventions, effective treatments for HFpEF have yet to be identified (Borlaug & Paulus, 2011).

Critically short telomeres, induced by breeding of multiple generations of mice lacking the catalytic subunit of telomerase *Terc*, lead to cardiac dysfunction and myocardial remodelling (Wong *et al*, 2008). However, replicative senescence-induced telomere shortening is unlikely to reflect normal physiological myocardial ageing, considering that the majority of CMs are post-mitotic, withdrawing from the cell cycle shortly after birth. While the heart has a limited potential for regeneration, CM turnover rates in both humans and mice are extremely low (Senyo *et al*, 2013; Bergmann *et al*, 2015; Richardson *et al*, 2015). We and others have reported that stress-induced telomere damage can also lead to telomere dysfunction and induce senescence (Fumagalli *et al*, 2012; Hewitt *et al*, 2012). Therefore, we investigated whether this phenomenon occurs in CMs and whether CM senescence contributes to myocardial remodelling during ageing.

Here, we demonstrate for the first time that a CM senescence-like phenotype is a feature of normal physiological human and murine ageing and provide a novel mechanistic model by which senescence occurs in rarely dividing/post-mitotic tissues. Persistent DNA damage foci, which co-localise with telomere regions, increase in cardiomyocytes with age independently of telomere length, telomerase activity or DNA replication and can be induced by mitochondrial dysfunction *in vitro* and *in vivo*. Global gene expression analysis of purified CMs, isolated from young and old mice, indicates activation of the classical senescence growth arrest pathways and of a distinct non-canonical SASP with functional effects including the potential to promote myofibroblast differentiation in fibroblasts and CM hypertrophy. Furthermore, specific induction of length-independent telomere dysfunction in CMs induces several senescence markers, including hypertrophy, which importantly is also associated with CM senescence *in vivo*. Finally, we demonstrate that suicide gene-mediated ablation of p16^Ink4a-expressing senescent cells and treatment with the senolytic drug, navitoclax, reduces CMs containing dysfunctional telomeres and attenuates cardiac hypertrophy and fibrosis in aged mice.

# Results

## Length-independent telomere damage in aged cardiomyocytes *in vivo*

Previously, we defined a novel mechanism of cellular senescence *via* the induction of irreparable telomere damage that occurs in the absence of telomere shortening (Hewitt *et al*, 2012). These data led us to hypothesise that this form of telomere damage could be the process by which senescence occurs in post-mitotic, or rarely dividing cells, which are not subject to proliferation-associated telomere shortening. We therefore investigated telomere dysfunction in adult mice throughout ageing by using dual Immuno-FISH to quantify co-localisation between DDR proteins γH2A.X, 53BP1 and telomeres, hereafter referred to as telomere-associated foci (TAF) in CMs (Fig 1A). CMs were identified by CM-specific markers α-actinin, troponin-C and the perinuclear protein PCM1 (Richardson, 2016; Richardson *et al*, 2015; Fig EV1A–C).

Mean TAF per CM and the total % of CMs containing TAF increased significantly with age in both mouse and human hearts (Figs 1A–C and EV1A). In contrast, the total number of γH2A.X foci did not change with age and were detected in almost all adult CMs irrespective of age (Fig 1D and E). Similar results were obtained when analysing co-localisation between DDR protein 53BP1 and telomeres in young and old mice (Fig EV1B). Interestingly, CMs from old animals contained a significantly higher number of TAF than other cardiac cell types (Fig EV1C), suggesting that telomere dysfunction is a predominant feature of CM ageing. To confirm further the localisation of a DDR at telomeres, we performed chromatin immunoprecipitation (ChIP) on cross-linked chromatin from heart tissue, using an antibody against γH2A.X followed by quantitative real-time PCR for telomeric repeats. We found enrichment of γH2A.X at telomeres in 30-month-old mice compared to 3-month-old mice (Fig 1F).

Although the majority of the adult CMs are considered post-mitotic, the adult heart retains limited potential for CM proliferation (Senyo *et al*, 2013; Bergmann *et al*, 2015; Richardson *et al*, 2015). We next investigated whether increased TAF was the outcome of telomere shortening driven by replication and/or impaired telomerase activity. Having first observed that telomerase activity was not affected by mouse age (Fig 1G), we observed a significant reduction in telomere FISH signal when comparing 4- to 30-month-old mice (Fig 1H). However, the median telomere FISH signal detected in old wild-type (30 months) mice was comparatively higher than that found in fourth-generation Terc$^{-/-}$ (G4) where critical telomere shortening has been shown to induce cardiac dysfunction (Wong *et al*, 2008; Fig 1I).

3D super-resolution stimulated emission depletion microscopy (STED) was used to more accurately detect telomere length by Q-FISH in CMs. STED resolution is typically 50 nm in XY and 150 nm in Z and overcomes the so-called diffraction limit of conventional confocal microscopy, which yields resolutions of ≥ 200 nm. Using

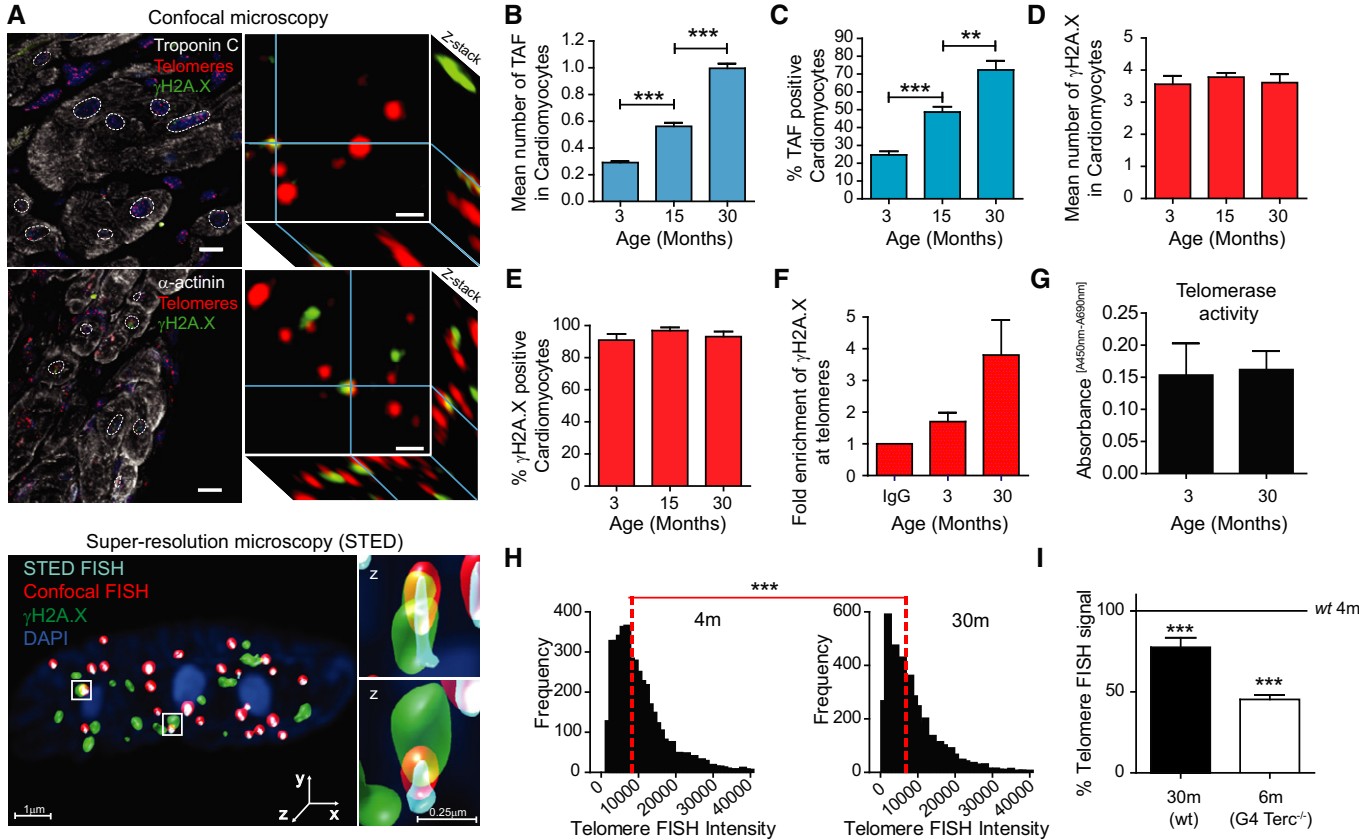

**Figure 1. Telomere dysfunction increases with age in mouse cardiomyocytes.**

A     Representative images of γH2AX immuno-FISH in troponin-C-positive/α-actinin-positive (top left and bottom left panels, respectively) mouse CMs (white—troponin-C/α-actinin; red—telo-FISH; green—γH2AX). Images are Z-projections of 0.1 μm stacks taken with a 100× oil objective. Scale bar: 20 μm. Right panels represent a single z-planes where co-localisation between a γH2AX foci and telomere was observed. Scale bars 1 μm. (Below) 3D reconstruction of Immuno-FISH using STED microscopy for γH2A.X and telomeres in a CM from 30-month-old mice. Scale bars as indicated and scale the same for both individual TAF examples using STED.

B, C     Mean number of TAF (B) and % of TAF-positive (C) α-actinin-positive CMs from C57BL/6 mice. Data are mean ± SEM of $n$ = 4–7 mice per age group. 100 α-actinin-positive CMs were quantified per mouse.

D, E     Mean number of total γH2A.X foci (D) and % of γH2A.X foci-positive nuclei (E) in α-actinin-positive CMs from C57BL/6 mice. Data are mean ± SEM of $n$ = 4–7 mice per age group. 100 α-actinin-positive CMs were quantified per mouse.

F     Fold enrichment of γH2AX at telomere repeats by real-time PCR. Graph represents fold enrichment of γH2AX at telomeric repeats between IgG control, 3- and 30-month-old whole mouse hearts. Data are mean ± SEM of $n$ = 3 mice per age group.

G     Quantitative PCR-ELISA TRAP assay comparing telomerase activity of 3- and 30-month-old C57BL/6 mice whole heart lysates. Data are mean ± SEM of $n$ = 4 mice per age group.

H     Histograms displaying distributions of telomere intensity in CMs analysed by 3D Q-FISH in young (4 months) and old (30 months) wild-type mice. Data are from $n$ = 3 mice. > 100 CMs were analysed per mouse.

I     % of telomere FISH signal loss in 30-month-old wild-type mice and late generation Terc$^{-/-}$ mice (6 months old) in comparison with 4-month-old wild-type mice. Data are from $n$ = 3 mice. > 100 CMs were analysed per mouse.

Data information: Statistical analysis was performed using one-way ANOVA (Holm–Sidak method) for multiple comparisons and Mann–Whitney test and two-tailed $t$-test for single comparisons. ***$P$ < 0.001; **$P$ < 0.01.

STED, we could identify an increased number of telomeres detected per CM, as well as decreased individual telomere volume when compared to conventional confocal microscopy (Fig EV1D and E, and Movie EV1). STED microscopy also revealed the existence of clusters of telomeres with varying lengths, which would otherwise be identified as single telomere signals by confocal microscopy (Fig EV1D).

We then proceeded to analyse telomere FISH intensities in telomeres co-localising with γH2A.X (TAF) and those not co-localising with γH2A.X (non-TAF) in aged mice using STED and found no significant differences (Fig EV1F and G). Similar observations were

made in CMs from aged human hearts (Fig EV1H). Analysis of telomere FISH intensities by STED in individual cardiomyocytes revealed that telomeres with low FISH intensity co-localising with γH2A.X are relatively rare (Fig EV1I). Together, our data suggest that length-independent telomere-associated DNA damage is unlikely to be due to the limitations of conventional confocal microscopy and that the shortest telomeres are not preferentially signalling a DDR.

While CM proliferation is negligible in adult humans and mice, it is possible that TAF may be the outcome of a small fraction of

dividing CMs. To ascertain further whether age-associated accumulation of TAF can occur in post-mitotic cells and is not exclusively linked to CM proliferation and turnover during ageing, we analysed our data in a mathematical model of a CM population over 27 months of a mouse lifespan. Our simulations reveal that, even considering relatively high estimates of cell division rates (4–15%), CM proliferation cannot account for the rate of the age-dependent increase in TAF. Rather, if we assume that proliferation plays no role in the TAF increase with age and that TAF are generated in non-dividing cells, there is a high degree of correlation between the model and our experimental data (Appendix Fig S1). In the $mdx^{4cv}/mTR^{G2}$ mouse model of telomere dysfunction, reduced expression of shelterin components is suggested to underlie increased telomere erosion in CMs (Mourkioti *et al*, 2013; Chang *et al*, 2016). To test whether uncapping of telomeres contributes to induction of a DDR at telomeres during ageing, we quantified the expression of shelterin components in CMs isolated from young and old wild-type mice and found no significant differences (Appendix Fig S2A). Similarly, we observed no significant difference in the abundance of TRF2 (an essential component of t-loop formation) in either TAF or non-TAF in human CMs *in vivo* (Appendix Fig S2B).

Together, these data support the notion that TAF increase with age in CMs and this occurs as a result of a process that is independent of cell proliferation can occur independently of telomere shortening and is not a result of overt alteration of telomere regulatory factors, such as shelterin components and telomerase. Having shown the phenomenon of telomere dysfunction occurring in CMs *in vivo*, we also found an age-dependent increase in TAF (but not non-TAF) in other post-mitotic cells, specifically in skeletal muscle myocytes and hippocampal neurons, which indicates the widespread nature of this phenomenon (Appendix Fig S3A–G).

**Telomere damage is persistent in cardiomyocytes**

Work by us and others previously demonstrated demonstrated that stress exposure leads to activation of a persistent DDR at telomeric regions (Fumagalli *et al*, 2012; Hewitt *et al*, 2012). To investigate the kinetics and nature of the DDR in CMs, we utilised different *in vitro* models. We first observed that exposure to X-ray radiation (10 Gy) resulted in both telomere-associated foci (TAF) and non-telomere-associated DNA damage foci (non-TAF) in mouse embryonic CMs positive for troponin-C and PCM1 (Fig 2A). However, only TAF were persistent, with non-TAF numbers being significantly reduced with time (Fig 2B).

Cultures of rat neonatal CMs and the myoblast cell line (H9C2) treated with stressors $H_2O_2$ and X-ray irradiation, respectively, showed that the number of TAF remained persistent over time (Fig 2C and D). Three days subsequently to X-ray irradiation, we monitored DDR in H9C2 cells using a 53BP1-GFP fusion protein (Passos *et al*, 2010) and time-lapse microscopy (Fig 2E). Over a 10-h time-course, the majority of individual DDR foci could be seen to be rapidly resolved with only approximately 20% of the original DNA damage foci persisting throughout the time-course (Fig 2F), similar to the % of TAF-positive cells for this time point (Fig 2D). To determine whether persistent TAF can also be induced in adult CMs *in vivo*, we exposed young mice to a whole-body dose of 2 Gy X-ray irradiation and measured the level of TAF in CMs after 11-month recovery. Both the mean number of TAF and % of CMs positive for TAF were significantly higher (Fig 2G). Altogether, these data suggest that the majority of genomic DNA damage in CMs is reparable and that only telomeric DNA damage is irreparable and persistent.

Double-strand breaks can arise in telomeres due to replication errors when replication forks encounter single-stranded breaks. In order to ascertain whether TAF could be induced in the absence of cell division, we pre-incubated H9C2 cells with EdU for 3 h and subjected them to either X-ray irradiation (10 Gy) or 10 μM $H_2O_2$. Following a 24-h recovery period (in the presence of EdU), the mean number of TAF was significantly increased in cells, which did not incorporate EdU (Appendix Fig S4A), indicating that TAF can be induced in the absence of DNA replication. Further, we isolated adult mouse CMs, which do not proliferate in culture, and exposed

**Figure 2. Stress-induced telomere-associated DNA damage is persistent in mouse embryonic cardiomyocytes, rat neonatal cardiomyocytes and H9C2 myoblasts.**

A  Representative images of mouse embryonic cardiomyocytes at days 0, 3, 5 and 10 days following 10 Gy X-irradiation. Left panels represent troponin-C-positive embryonic cardiomyocytes (troponin-C—magenta; DAPI—light blue). Middle panels display γH2AX foci (green) and telomeres (red) in Z-projections of 0.1 μm slices, with white arrows indicating co-localisation. Co-localising foci are amplified in the right-hand panels (amplified images represent a single z-planes where co-localisation was observed). Scale bars represent 10 μm. Scale bars in single-plane images 500 nm.

B  (Left) Mean number of both TAF and non-TAF in troponin I-positive mouse embryonic cardiomyocytes at days 0, 3, 5 and 10 following 10 Gy X-irradiation. Data are mean ± SEM of *n* = 3 independent experiments; 30–50 troponin-positive cardiomyocytes were analysed per experiment. (Right) Mean percentage of γH2AX foci co-localising with telomeres (% TAF) in troponin-C-positive mouse embryonic cardiomyocytes at days 0, 3, 5 and 10 following 10 Gy X-irradiation. Statistical analysis performed using one-way ANOVA (Holm–Sidak method); *P* < 0.05. Significant differences were found for mean number of non-TAF, but not for mean number of TAF.

C  Mean number of both TAF and non-TAF in neonatal rat cardiomyocytes at days 0, 3, 5, 10 days following treatment for 24 h with $H_2O_2$. Data are mean ± SEM of *n* = 3. > 50 cells were quantified per condition. Statistical analysis performed using one-way ANOVA (Holm–Sidak method); **P* < 0.01.

D  (Left) Representative images of γH2AX immuno-FISH in H9C2 myoblasts 3 days 10 Gy X-irradiation (red—telo-FISH; green—γH2AX). White arrows identify areas shown in higher magnification panels. (Right) Mean number of both TAF and non-TAF in H9C2 myoblasts at days 3 and 5 following 10 Gy X-irradiation. Data are mean ± SEM of *n* = 3. > 50 cells were quantified per condition. Statistical analysis performed using one-way ANOVA (Holm–Sidak method); ***P* < 0.001. Scale bars represent 1 μm. Scale bars in single-plane images 500 nm.

E  Representative time-lapse images of H9C2 rat cardiomyoblasts expressing AcGFP-53BP1 from 3 days after 10 Gy irradiation at the indicated times (min). Images are maximum intensity projections with a 6.7 μm focal depth. Scale bar represents 1 μm.

F  Kaplan–Meier survival curves for AcGFP-53BP1c DDR foci in H9C2 cells 3 days after 10 Gy irradiation at 10-min intervals for 24 h. > 500 foci from 10 cells were tracked per condition. Gehan–Breslow test was used, *P* < 0.001.

G  Schematic illustration showing 1-month-old C57BL/6 mice treated with 2 Gy whole-body X-irradiation, followed by a recovery period of 11 months before culling at 12 months of age. Mean number of TAF in α-actinin-positive cardiomyocytes. Data are mean ± SEM of *n* = 3 mice per group. Ninety α-actinin-positive cardiomyocytes were quantified per condition. Statistical analysis performed using two-tailed *t*-test; *P* < 0.05.

   

them to 5 μM $H_2O_2$. We observed that the mean number of TAF was significantly increased after treatment (Appendix Fig S4B). In addition, we did not find evidence that the shortest telomeres were specifically targeted by stressors. In fact, EdU-negative irradiated H9C2 cells had a slightly higher median telomere length in damaged telomeres (TAF) compared to telomeres that did not co-localise with γH2A.X (non-TAF). However, such a difference was not found in adult murine CMs treated with $H_2O_2$ (Appendix Fig S4C).

## Telomere-specific DNA damage drives a senescent-like phenotype

Having demonstrated that TAF increase with age and can be induced due to stress, we then tested whether telomere-specific damage could directly drive a senescent-like phenotype in the CM lineage. Rat neonatal CMs were transfected with an endonuclease that specially targets telomeres [a TRF1-FokI fusion protein and a

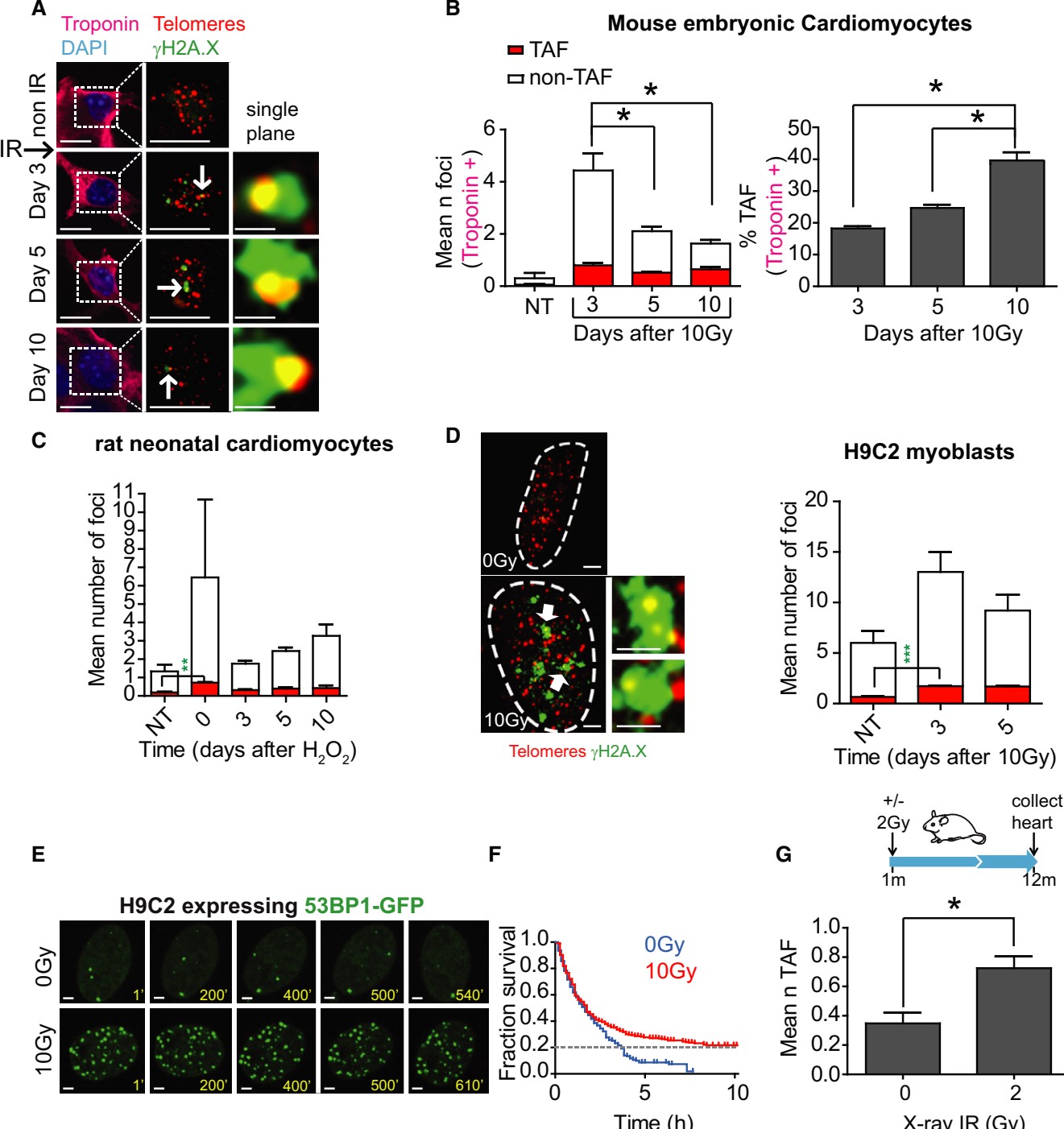

**Figure 2.**

TRF1-Fok1D450 inactive mutant (Dilley *et al*, 2016)]. In culture, we found that rat neonatal CMs had very low proliferation rates with 5% of cells incorporating EdU over a period of 24 h. Following 5 days of culture, CMs expressing TRF1-FokI showed numerous DNA damage foci with the significant majority of damage co-localising with telomeres (Fig 3A–C). Telomeres were targeted by the endonuclease irrespective of their length (Fig 3D). *De novo* TAF formation induced a senescent phenotype in CMs characterised, in addition to TAF, by increased SA-β-Gal activity and upregulation of the cyclin-dependent kinase inhibitor p21$^{CIP}$ (Fig 3E and F), as well as increased cellular hypertrophy (Fig 3G). Similar results were found using the H9C2 myoblasts (Fig EV2A–E). Additionally, we used the AC10 cell line derived from adult human ventricular CM (Davidson *et al*, 2005) stably expressing an inducible form of the TRF1-FokI construct. These cells express CM-specific transcription factors and contractile proteins and, while proliferative in complete media, are induced to terminal differentiation and exit the cell cycle under mitogen-depleted conditions. We induced TRF1-FokI expression in differentiated AC10 cells by exposure to doxycycline (DOX) for 6 h and allowed the cells to recover for an additional 36 h in the absence of DOX. Consistent with the idea that telomere damage is irreparable in differentiated CMs, we found that expression of TRF1-FokI significantly increased the mean number of TAF and this number persisted for at least 36 h (Fig EV2F and G). Additionally, expression of TRF1-FokI resulted in increased SA-β-Gal activity (Fig EV2H). In contrast, if similar numbers of non-telomeric DNA damage foci were induced using the homing endonuclease I-PpoI in neonatal cardiomyocytes, we observed that most DNA damage is repaired within 4 days, without induction of senescent markers or increased cell size (Fig EV2I–K).

## Aged cardiomyocytes activate senescence pathways but not a typical SASP

Within the heart, CMs comprise approximately only 30% of the total cells (Nag & Zak, 1979). To overcome this heterogeneity and obtain an accurate representation of the CM transcriptome during ageing, we devised a new method for isolation and enrichment of CMs. In short, we conducted a retrograde *Langendorff* perfusion for dissociation of cardiomyocytes, followed by removal of CD31$^+$/CD45$^+$/ScaI$^+$ interstitial cells via magnetic bead sorting (Fig 4A). This method allowed us to obtain a highly enriched cardiomyocyte population (Fig EV3A). RT–PCR quantification of mRNAs encoding the cyclin-dependent kinase inhibitors p16$^{Ink4a}$, p21$^{CIP}$ and p15$^{Ink4b}$ in 3- and 20-month-old animals demonstrated an age-dependent increase in expression of all three genes (Fig 4B). Immunohistochemistry on tissue sections from ageing mice validated the increase of p21$^{CIP}$ at the protein level, specifically in CMs (Fig 4C). Furthermore, we detected increased activity of SA-β-Gal in old mice (Fig 4D). While SA-β-Gal positivity was rare, we could detect it in CMs but no other cell types from old mice. By centromere-FISH in CMs, we also observed an age-dependent increase of senescence-associated distension of satellites (SADS), a marker of senescence (Swanson *et al*, 2013; Fig 4E). Global transcription analysis of purified CM populations by RNA-sequencing revealed 416 differentially expressed genes between young (3 months) and old (20 months; Fig EV3B) mice. Principal component analysis revealed that both cohorts separate well with

the first and second components, accounting for 80.1 and 7.8% of the cumulative variance across the data set (Fig EV3C). Consistent with rare CM proliferation *in vivo*, we did not observe any significant changes in the expression of proliferation genes (Fig EV3D). However, genes involved in the regulation of myofilaments, contraction and CM hypertrophy were enriched in old mice (Myh7, Capn3, Ankrd1, Myom3, Myot, Mybpc2). Confirmatory qRT–PCR showed age-dependent increase in hypertrophy genes Myh7 and Acta1, but not Anf (Fig EV3E and F). Interestingly, pro-inflammatory genes associated with the SASP (Coppé *et al*, 2008), such as Il-1α, Il-1β, Il-6, Cxcl1 and Cxcl2, were not differentially expressed between CMs from young and aged animals (Fig 4F). To determine whether SASP factors were secreted by CMs, we collected conditioned medium from isolated CMs from young and old mice. In accordance with the RNA-seq results, we did not find any significant differences in the levels of secreted proteins using a cytokine array (Fig 4F).

Our data are in contrast with a previous report that shows increased expression of SASP components in murine heart with age using whole heart homogenates (Ock *et al*, 2016). We therefore compared the expression of previously reported SASP components in non-purified and purified CM populations. Consistently, we found that following a simple *Langendorff* digestion that collects a heterogeneous population of CMs and stromal cells, we found significant differences in expression of SASP components such as Il-6 and Cxcl1 between young and old mice (Appendix Fig S5A). However, the population of purified CMs demonstrated no such differences, suggesting that cell types other than CMs could explain previous observations (Appendix Fig S5A). Interestingly, RNA sequencing led to the identification of three secreted proteins, not commonly categorised as SASP components, which were confirmed to be significantly increased at the mRNA level in aged purified CMs: Edn3, Tgfb2 and Gdf15 (Fig 5A). Of these, only Edn3 was increased exclusively in aged CMs, but neither in cardiac stromal cells nor in other organs, suggesting that increased Edn3 levels in plasma is a good indicator of CM ageing (Appendix Fig S5B–D). The SASP has been shown to impact on proliferation of neighbouring cells, ultimately inducing senescence (Acosta *et al*, 2013). Consistently, we found that conditioned medium isolated from old adult CMs reduced proliferation of neonatal fibroblasts (measured by EdU incorporation) and increased the expression of α-smooth muscle actin (α-SMA), an indicator of myofibroblast activation (Fig 5B–E). We then examined in more detail the role of the newly identified CM senescence-associated proteins in terms of their ability to induce bystander effects, particularly their role in fibrosis, proliferation and hypertrophy. We found that Edn3, Tgfb2 and Gdf15 induced expression of α-SMA in fibroblasts; however, only Tgfb2 reduced cellular proliferation as measured by EdU incorporation (Fig 5F–H). Furthermore, treatment of neonatal rat CMs with Edn3 or Tgfb2, but not Gdf15, resulted in a significant increase in the cell surface area, supporting their involvement in the induction of hypertrophy (Fig 5I). Consistent with previous reports indicating that the SASP can induce senescence in neighbouring cells (Acosta *et al*, 2013), we found that conditioned medium from old adult CMs was able to induce an increase in SA-β-Gal and a reduction in EdU incorporation in fibroblasts. No significant increase in DNA damage foci was observed in fibroblasts (Appendix Fig S5E–H).

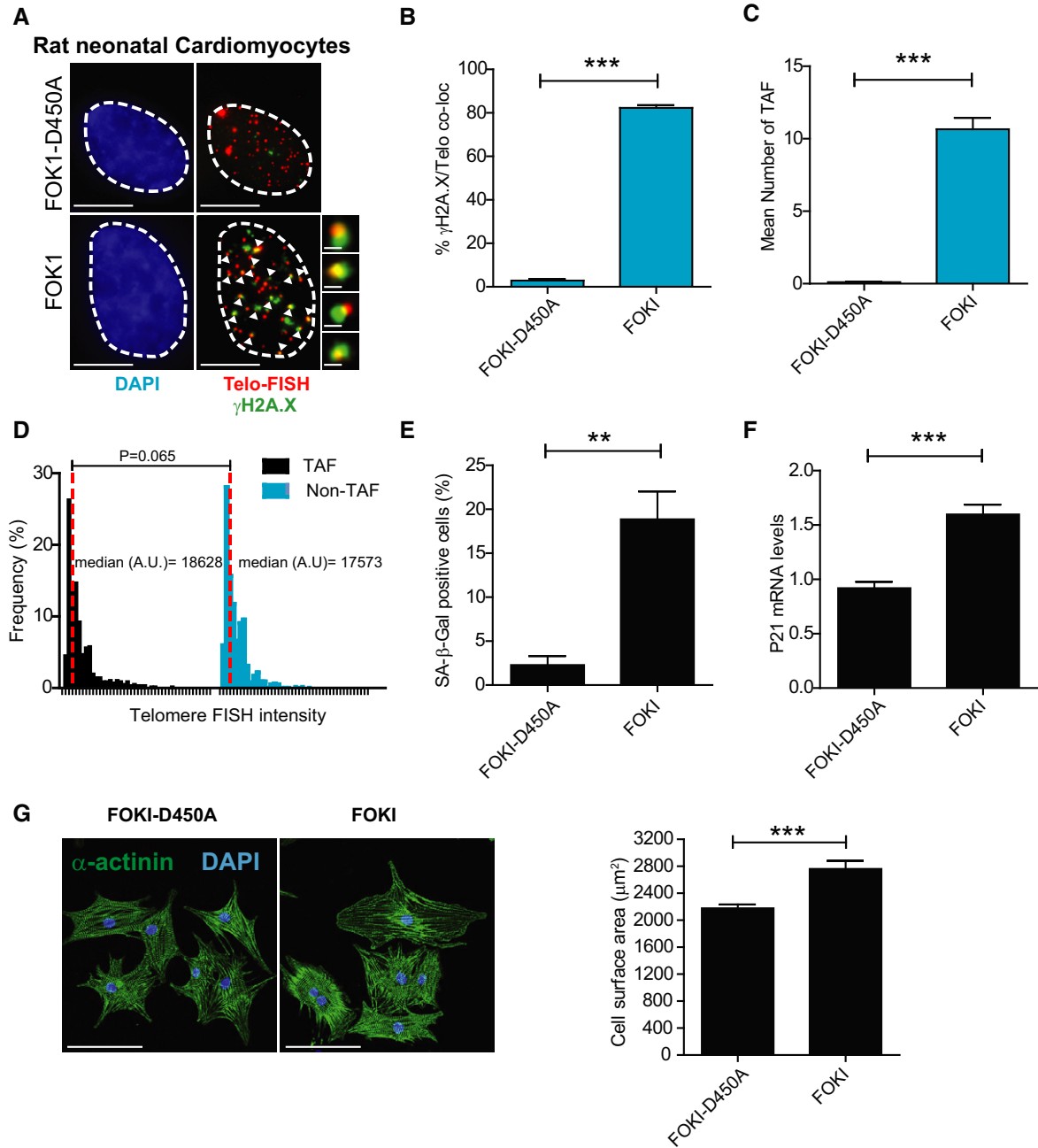

**Figure 3. TRF1-FokI fusion protein induces telomere-specific double-strand breaks, senescence and hypertrophy in rat neonatal cardiomyocytes.**

A    Representative images of rat neonatal CMs 4 days following transfection with a FLAG-tagged TRF1-FokI-D450A (top row) or TRF1-FokI (middle and bottom row) fusion protein (cell treatments the same for all subsequent panels in Figure; red—telo-FISH; green—γH2A.X). Images are z-projections of 0.1 μm stacks taken with 100× objective. White arrows indicate co-localisation between telomeres and γH2A.X, with co-localising foci amplified in the right panels (taken from single z-planes where co-localisation was found). Scale bar represents 3.5 μm. Scale bar in magnified images showing individual co-localisation 500 nm.

B, C    % of γH2A.X foci co-localising with telomeres (B) and mean number of telomere-associated foci (TAF) (C) in FLAG-tagged TRF1-FokI-D450A- and TRF1-FokI-expressing CMs. Data are mean ± SEM of *n* = 4 independent experiments. > 50 cells were analysed per condition. Statistical analysis was performed by two-tailed *t*-test ***$P$ < 0.001.

D    Histograms displaying telomere intensity for telomeres co-localising or not co-localising with γH2AX foci. Red dotted lines represent median. Mann–Whitney test shows no significant difference in telomere intensity between TAF and non-TAF.

E    Mean % of FLAG-labelled CMs positive for SA-β-Gal activity. Data are mean ± SEM of *n* = 3 independent experiments. > 100 cells were quantified per condition. Statistical analysis performed using two-tailed *t*-test; **$P$ < 0.01.

F    Expression of p21 mRNA (as a function of β-actin and Gapdh) by real-time PCR in TRF1-FokI-D450A and TRF1-FokI-expressing CMs. Data are mean ± SEM of *n* = 6 independent experiments. Statistical analysis performed using two-tailed *t*-test; ***$P$ < 0.001.

G    Mean ± SEM cell surface area (μm²) of FLAG-labelled CMs expressing TRF1-FokI-D450A and TRF1-FokI. Statistical analysis performed using two-tailed *t*-test; ***$P$ < 0.001. Scale bar represents 50 μm.

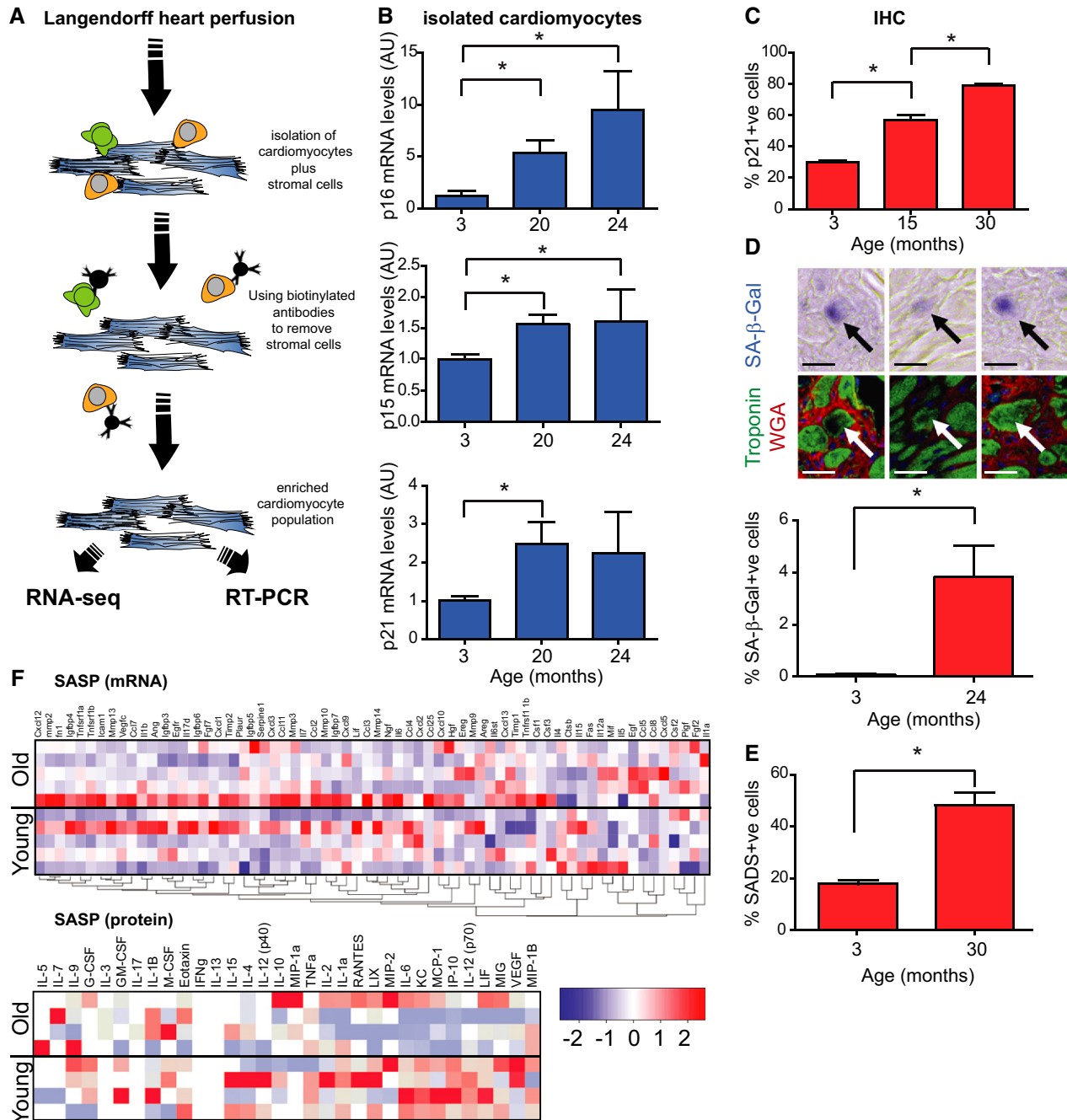

**Figure 4. Aged cardiomyocytes activate senescence pathways but not a typical SASP.**

A   Scheme illustrating CM isolation and purification procedure.

B   Real-time PCR gene expression analysis in isolated mouse CMs from C57BL/6 mice. Data are mean ± SEM of $n$ = 3–4 per age group. Statistical analysis performed by one-way ANOVA (Holm–Sidak method); *$P$ < 0.05.

C   Mean % of p21-positive CM nuclei from 3-, 15- and 30-month-old C57BL/6 mice by immunohistochemistry (IHC). Data are mean ± SEM of $n$ = 4 per age group. > 100 CMs were quantified per age group. Statistical analysis performed using one-way ANOVA (Holm–Sidak method); *$P$ < 0.05.

D   Mean % of 3- and 24-month-old mouse CMs staining positive for SA-β-Gal *in vivo* with representative images above (blue—SA-β-Gal; green—troponin-C; red—WGA). Black arrows indicate SA-β-Gal expression in a troponin-C-expressing CM. Statistical analysis performed using two-tailed *t*-test; *$P$ < 0.05. Data are mean ± SEM from the analysis of > 500 CMs per mouse, four mice per age group. Scale bar 20 μm.

E   Mean % of SADS-positive CM nuclei from 3- and 30-month-old mouse CMs positive for SADS *in vivo*, as detected by centromere-FISH. Data are mean ± SEM of $n$ = 4 per age group. > 200 CMs per mouse were quantified. Statistical analysis performed using two-tailed *t*-test; *$P$ < 0.05.

F   SASP heatmap: Pearson correlation clustered heatmap showing a curated list of known SASP genes (top panel) or a selection of secreted SASP proteins (bottom panel) in young (3 months) and old (20 months) mouse CMs ($n$ = 5 per age group). The colour intensity represents column Z-score, where red indicates highly and blue lowly expressed.

We conclude that aged CMs *in vivo* activate a number of senescence effector pathways, including hypertrophy and a non-typical SASP, which may mediate both autocrine and paracrine effects.

### Mitochondrial ROS drive TAF in mouse cardiomyocytes *in vivo*

Mitochondrial dysfunction has been described as both a driver and consequence of cellular senescence (Passos *et al*, 2007, 2010; Correia-Melo *et al*, 2016). Gene Set Enrichment Analysis (GSEA) of RNA-seq data revealed that the most negatively enriched Gene ontology term in old CMs is the mitochondrial inner membrane (Fig EV4A). Consistently, in CMs from old mice we observed an overall decline in expression of most mitochondrial genes, particularly those genes involved in the electron transport chain (ETC; Fig 6A) and mitochondrial ultrastructural defects by transmission electron microscopy (Fig EV4B).

We then speculated whether mitochondrial ROS could be a driver of telomere dysfunction in CMs *in vivo*. Previous data indicate that mitochondrial ROS can drive telomere shortening *in vitro* (Passos *et al*, 2007) and that telomere regions are particularly sensitive to oxidative damage (Hewitt *et al*, 2012). Consistent with this hypothesis, we found increased mRNA expression of the pro-oxidant enzyme monoamine oxidase A (MAO-A) and decreased expression of antioxidant enzymes mitochondrial MnSOD and catalase in isolated CMs from old mice (Fig 6B). Furthermore, we found increased lipid peroxidation marker, 4-HNE, and DNA oxidation marker, 8-oxodG, in hearts with age (Fig 6C). To address whether these age-associated changes are causal in TAF induction, we utilised a model of CM-specific overexpression of MAO-A (MHC-MAO-A tg), which displayed enhanced levels of mitochondrial ROS specifically in CMs (Villeneuve *et al*, 2013). In this transgenic mouse, both the mean number and % of CMs positive for TAF were significantly increased compared to age-matched controls (Fig 6D). Critically, when transgenic animals were treated with the antioxidant N-acetyl cysteine (NAC), both the increase in TAF (Fig 6D) and CM hypertrophy were significantly rescued (Fig EV4C). Complementary studies in old MnSOD$^{+/-}$ and Catalase$^{-/-}$ mice also revealed that these had a higher number of TAF than age-matched controls (Fig 6E and F). Furthermore, we found a significant increase in TAF in CMs from Polg$^{mut/mut}$ mice, a model of accelerated ageing due to mitochondrial dysfunction (Trifunovic *et al*, 2004; Fig 6G). Increased telomere dysfunction in these mice

was associated with decreased expression of mitochondrial proteins, increased expression of p21, and cardiac hypertrophy (Fig EV4D). Finally, treatment of isolated adult CMs with rotenone, a mitochondrial complex I inhibitor, induced TAF, which could be rescued by treatment with the antioxidant NAC (Fig 6H).

In summary, CMs from aged heart exhibit downregulation of mitochondrial inner membrane and ETC. genes, and this is associated with indicators of increased ROS metabolism that is causative of telomere-associated DNA damage.

### Senescent-cell clearance reduces cardiac hypertrophy and fibrosis

Myocardial remodelling occurs with ageing. Furthermore, it accompanies diseases such as ischaemia, hypertension, valvular disease and heart failure. In order to assay cardiac function with ageing in mice, we used magnetic resonance imaging (MRI). We found no significant difference in systolic cardiac function (ejection fraction) with age. However, we observed an increased mean left ventricle mass (mean of diastolic and systolic mass) indicative of hypertrophy and increased ventricle wall rigidity, symptomatic of a decline in diastolic function, both of which are characteristics of HFpEF patients (Borlaug, 2014) and aged mice (Dai *et al*, 2012; Fig 7A).

Previously, we showed that specific induction of TAF was associated with increased cell size in rat neonatal CMs *in vitro* and that CM senescence is associated with hypertrophy *in vivo*. Consistent with a role for TAF-induced senescence in age-dependent cardiac hypertrophy, we found that larger CMs from old mice had generally higher TAF frequencies (Fig 7B). In order to determine whether there is a causal relationship between senescence and cardiac hypertrophy, we used the INK-ATTAC mouse model, in which a small molecule, AP20187 (AP), induces apoptosis through dimerisation of FKBP-fused Casp8. Using this model, it has previously been shown that clearance of p16-expressing cells improves multiple parameters of physical health and function within ageing mice (Baker *et al*, 2011, 2016; Farr *et al*, 2017; Ogrodnik *et al*, 2017).

In order to establish if elimination of p16$^{Ink4a}$-positive cells reduced TAF in CMs, we aged INK-ATTAC mice until they were 27 months old and treated them with AP for 2 months (Fig 7C). We confirmed by RNA *in situ* hybridisation that both p16$^{Ink4a}$- and eGFP-positive cells were significantly reduced in hearts following

---

**Figure 5. Cardiomyocyte SASP induces fibrosis and reduces proliferation in fibroblasts and induces hypertrophy in cardiomyocytes.**

A    (Above) RNA sequencing of purified CMs from four mice per age group reveals age-dependent increased expression of three secreted proteins Edn3, Tgfb2 and Gdf15; the colour intensity represents column Z-score, where red indicates highly and blue lowly expressed. (Below) mRNA expression of Edn3, Tgfb2 and Gdf15 was independently validated by RT–PCR in young and old isolated adult CMs. Data are mean ± SEM of *n* = 5 mice per group.

B    CMs were isolated by *Langendorff* heart perfusion, purified and cultured for 48 h. Conditioned medium (CM) was collected and added to cultures of neonatal fibroblasts in the presence of 10 μM EdU.

C    Representative images of immunofluorescent staining against α-SMA and EdU in neonatal fibroblasts cultured in the presence of CM from young and old CMs.

D, E    Quantification of % of EdU incorporation (D) and % of a-SMA-positive cells (E) in neonatal fibroblasts after treatment for 48 h with CM from young and old CM. Data are mean ± SEM of *n* = 3–4 mice per age group; > 200 cells were quantified per condition.

F    Representative micrographs of neonatal fibroblasts and CMs treated with recombinant proteins: Tgfb2, Edn3 and Gdf15 for 48 h and immunostained against α-SMA and EdU (fibroblasts) and α-actinin (CMs).

G–I    Quantification of α-SMA- (G) and EdU-positive neonatal fibroblasts (H) and surface area (μm²) (I) in neonatal CMs following treatment with the indicated recombinant proteins. Data are mean ± SEM of *n* = 3–4 independent experiments.

Data information: Asterisks denote statistical significance with one-way ANOVA (multiple comparisons) (G, H, I) or two-tailed *t*-test (A, D, E). ***$P$ < 0.001; **$P$ < 0.01; *$P$ < 0.05. For all panels, scale bars represent 50 μm.

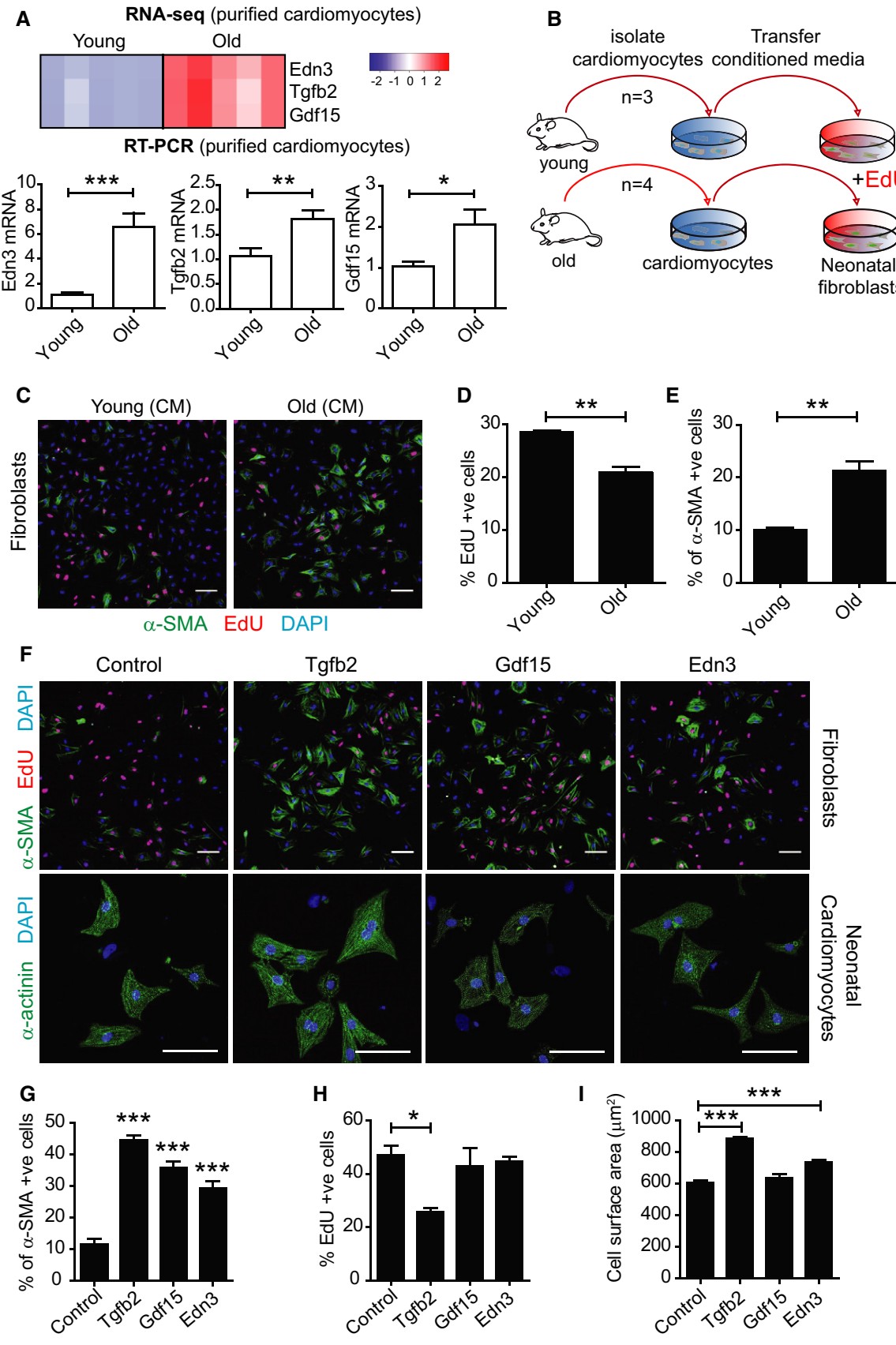

**Figure 5.**

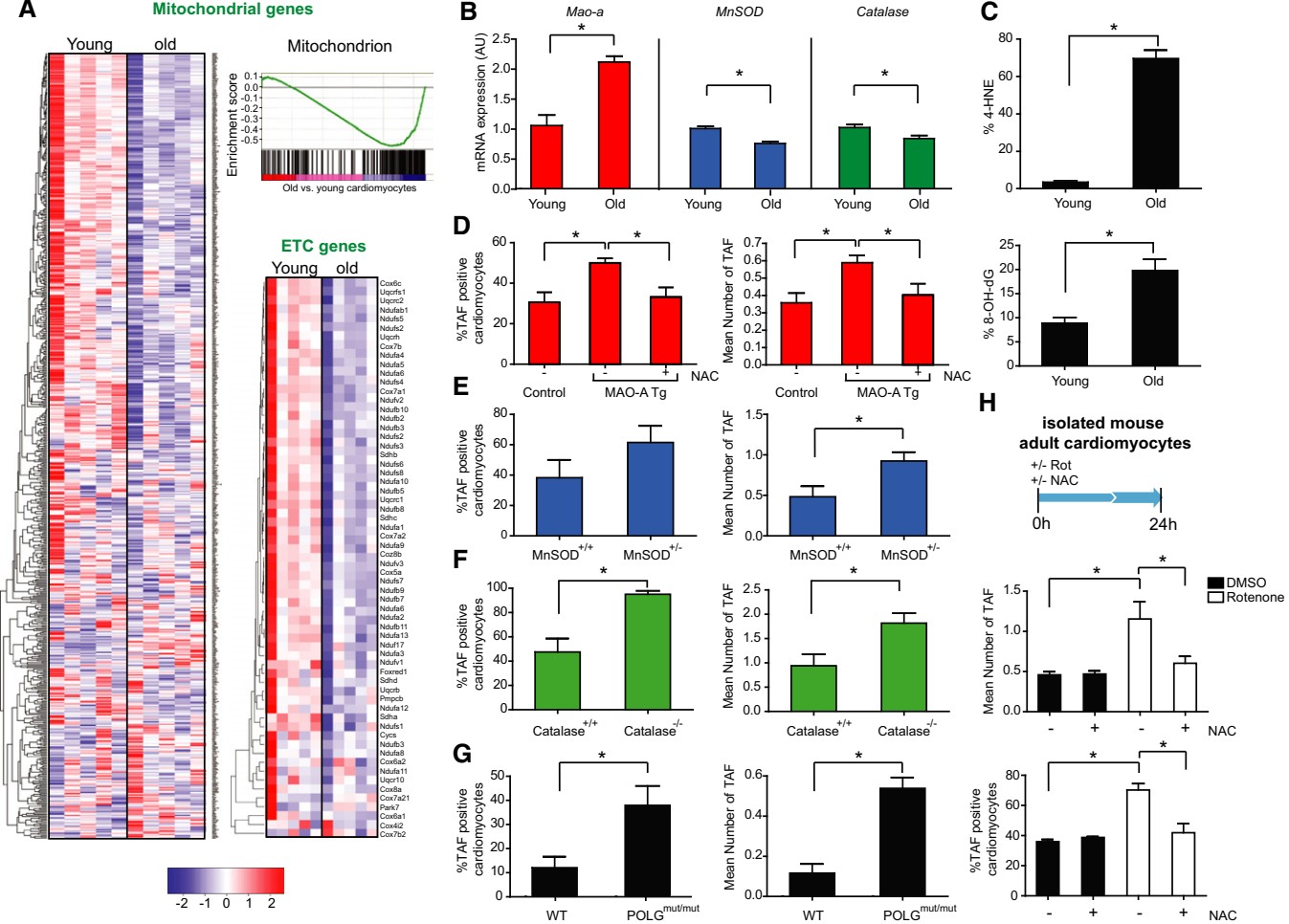

**Figure 6.** Mitochondrial dysfunction is a feature of cardiomyocyte senescence and drives TAF in mouse cardiomyocytes *in vivo*.

A  Mito and ETC genes with GSEA analysis: Clustered heatmap showing all genes associated with the "Mitochondrion" GO term in young and old, mouse CMs as observed by the GSEA pre-ranked list enrichment analysis (normalised enrichment score: −1.70; FDR *q*-value < 0.05). Alongside this is a column clustered heatmap displaying a list of genes from the electron transport chain (ETC) GO ontology. In both instances, genes are by column and samples by row with the colour intensity representing column *Z*-score, where red indicates highly and blue lowly expressed.

B  Real-time PCR gene expression analysis of MAO-A, MnSOD and catalase in isolated mouse CMs from young (3 months) and old (20 months) mice. Data are mean ± SEM of *n* = 4–5 per age group. Asterisks denote a statistical significance at $P < 0.05$ using two-tailed *t*-test.

C  Mean % of 4-HNE- (top) or 8-oxodG-positive (bottom) CMs from 3 month (young) or 30 month (old) mice. Data are mean ± SEM of *n* = 4 per age group. 100 CMs were quantified per age group. Asterisks denote a statistical significance at $P < 0.05$ using two-tailed *t*-test.

D–G  Mean % of TAF-positive nuclei (left graphs) or mean % of TAF (right graphs) in wild-type (control) compared to MAO-A transgenic mice with or without drinking water supplemented with 1.5 g/kg/day NAC from the age of 4–24 weeks, MnSOD$^{+/+}$ vs MnSOD$^{−/+}$, Catalase$^{+/+}$ vs Catalase$^{−/−}$, WT vs POLG double-mutant mice. Data are mean ± SEM of *n* = 3–4 per group. > 100 CMs were quantified per age group. Statistical analysis was performed using two-tailed *t*-test (E–G) or one-way ANOVA (for multiple comparisons) (D); *$P < 0.05$.

H  Scheme depicting isolated mouse adult CMs isolated from four animals were treated with or without 100 nM rotenone either in the presence of 5 mM NAC or vehicle control (pre-treated for 30 min before rotenone treatment), for 24 h before fixation. Mean number of TAF (top graph) and mean % of TAF-positive nuclei (bottom graph). Data are mean ± SEM from four separate CM cultures isolated from 3-month-old mice. Fifty CMs were quantified per condition. Asterisks denote a statistical significance at $P < 0.05$ using one-way ANOVA (Holm–Sidak method).

AP treatment (Fig 7D). Additionally, we found that TAF in CMs were significantly reduced (Fig 7E); however, we failed to detect any differences in telomere FISH signals between INK-ATTAC mice with or without clearance of p16$^{Ink4a}$-positive cells (Fig EV5A). Consistent with a role for CM senescence in age-dependent myocardial remodelling, we found that AP treatment significantly reduced the average cross-sectional area of CMs (Fig 7F), and decreased the percentage area of fibrosis, without any change in heart function as measured by % ejection fraction (Figs 7G and EV5B and C). Similar results were observed in a model of cardiac hypertrophy induced by thoracic irradiation. We found that TAF induced by thoracic irradiation in INK-ATTAC mice were restored to the levels found in sham-irradiated mice following AP treatment (Appendix Fig S6A–C). Similarly, we found that thoracic irradiation resulted in CM hypertrophy and this was completely rescued by elimination of p16$^{Ink4a}$-positive cells (Appendix Fig S6D).

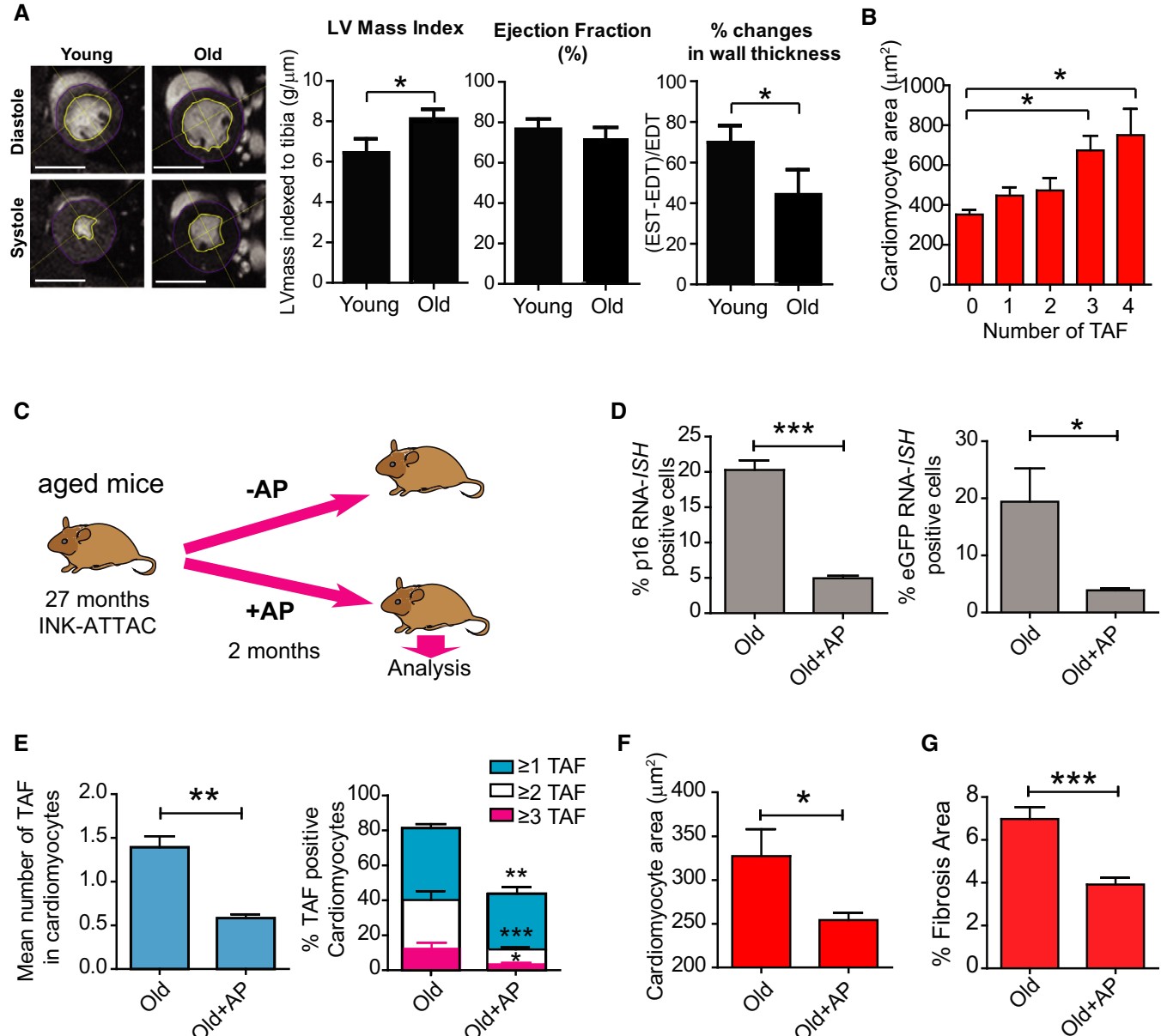

**Figure 7. Genetic clearance of p16-positive cells in aged mice reduces cardiomyocyte senescence, cardiac hypertrophy and fibrosis.**

A   Examples of individual short-axis cine-MR images of 3-month-old or 20-month-old mouse hearts. Ejection fraction and LV thickness were calculated based on manual measurements of left ventricle epicardial and endocardial borders. % change in wall thickening calculation based on wall thickness at the four points indicated. Measurements were made in all cine slices at end diastole and end systole. Graphs representing data obtained from MRI analysis of > 7 animals per age group. Data are mean ± SEM. Asterisks denote a statistical significance at P < 0.05 using Mann–Whitney U-test. Scale bars represent 5 mm.

B   Comparison between mean number of TAF per CM and CM area in 22-month-old animals. Data are mean ± SEM of n = 4. > 100 CMs were quantified per mouse. Statistical analysis performed using one-way ANOVA (Holm–Sidak method); *P < 0.05.

C   Scheme depicting experimental design for (D–F): 27-month-old INK-ATTAC mice were treated 4 times with AP20187 (or vehicle), 3 days in a row, every 2 weeks (2-month-long treatment in total) and were sacrificed afterwards for analysis.

D   Comparison between the % of p16- or eGFP-positive CMs by RNA in situ hybridisation per plane in INK-ATTAC mice (28–29 months old) treated with vehicle or AP20187. Data are mean ± SEM of n = 5 per age group. 100 CMs were analysed per mouse. Asterisks denote a statistical significance at *P < 0.05 or ***P < 0.001 using two-tailed t-test.

E   Mean number of TAF (left graph) and mean % of TAF-positive nuclei (right graph) in CMs. Data are mean ± SEM of n = 6 per age group. 100 CMs were analysed per mouse. ***P < 0.001; **P < 0.01; *P < 0.05.

F   Mean CM area μm². Data are mean ± SEM of n = 6 per age group, > 150 CMs analysed per mouse. Asterisks denote a statistical significance at *P < 0.05 using two-tailed t-test.

G   % of fibrotic area evaluated by Sirius Red staining. Data are mean ± SEM of n = 6 per age group. Asterisks denote a statistical significance using Mann–Whitney test. ***P < 0.001.

To investigate further the therapeutic impact of targeting senescent cells to counteract cardiac ageing, we treated aged wild-type mice with the previously described senolytic drug, ABT263 (navitoclax; Zhu *et al*, 2016), or vehicle intermittently for 2 weeks (Fig 8A). We found that navitoclax reduced telomere dysfunction in cardiomyocytes without affecting telomere length (similar to what was observed in aged INK-ATTAC mice following AP treatment; Figs 8B and C, and EV5D). Consistent with its senolytic properties, we found that navitoclax selectively killed senescent H9C2 CMs *in vitro* (Fig EV5E and F).

Similarly, to genetic clearance of p16$^{Ink4a}$ cells in INK-ATTAC mice, we found that navitoclax significantly reduced hypertrophy and fibrosis in aged wild-type mice (Fig 8C and D). However, navitoclax had no significant impact on cardiac function, LV mass and ventricle wall rigidity (Fig 8E–G). The decrease in mean CM size without significant changes in LV mass suggested a compensatory increase in overall CM number. Supporting *de novo* CM proliferation, we observed that frequency distribution analyses of CM cross-sectional area suggested that the decrease in mean CM area following navitoclax treatment is a function of both an elimination of the largest CMs, presumably as these are senescent, and the appearance of a "new" population of small CMs (Fig EV5G). This phenotype has previously been associated with CM regeneration (Waring *et al*, 2014). To further investigate *de novo* CM regeneration following navitoclax treatment, we injected EdU in 24-month-old mice for 7 days and quantified EdU-positive cells in combination with CM markers TropC and PCM1 and cell membrane marker wheat germ agglutinin (WGA). We found significantly more EdU-labelled CMs in the navitoclax-treated animals than vehicle controls (0.23% vs 0.07%; Fig 8H–J). As CMs may undergo karyokinesis in the absence of cytokinesis resulting in bi- or multi-nucleation, EdU alone is not a marker of cell division. As such, we evaluated the nucleation state of EdU-labelled CMs by analysing 3D images in 40-μm-thick sections as previously described (Mollova *et al*, 2013). Our data indicate senescent-cell clearance is accompanied by a significant increase of both multi- and mono-nucleated CMs (Fig 8H–J). Each image throughout the z-series was examined to confirm nucleation state (Movies EV2 and EV3). However, < 25% of all EdU incorporation resulted in multi-nucleated CMs. We next evaluated the percentage of CMs positive for proliferation marker Ki-67 and found a similar increase following navitoclax treatment and in aged INK-ATTAC mice treated with AP (Figs EV5H and 8K). Expression of Aurora B, a component of the contractile ring required for cytoplasmic separation during cell division, was only observed in CMs from hearts where senescent cells had been cleared, where it was located in mid-body between nuclei, indicative of cytokinesis (Hesse *et al*, 2018; Fig 8H). Altogether, our data support that senescent CMs are involved in age-associated cardiac hypertrophy and fibrosis and that their clearance may induce a compensatory CM regeneration.

## Discussion

During ageing and despite low level of proliferation, we observe enrichment of DDR proteins at telomere regions, notoriously known for their inefficient repair capacity, fuelled by the actions of telomere binding proteins. TRF2 and its binding partner RAP1 have been

shown to prevent NHEJ-dependent telomeric DNA fusions by inhibiting DNA-PK- and ligase IV-mediated end-joining (Bae & Baumann, 2007). In contrast, homologous recombination can repair telomere-induced DNA damage; however, this process is restricted to proliferating cells undergoing S-phase and therefore not likely to be relevant for post-mitotic CMs (Mao *et al*, 2016). As such, within CMs, a cocktail of age-dependent mitochondrial dysfunction and oxidative stress, coupled with limited CM turnover, fuels the occurrence of irreparable telomere-associated damage, which may instigate a senescent-like phenotype (Appendix Fig S7).

While our data indicate that mitochondrial dysfunction and ROS can induce CM senescence, it remains possible that the observed age-dependent changes in mitochondrial gene expression and morphology are a consequence of telomere dysfunction. In mouse models of accelerated telomere shortening, critically short telomeres have been shown to repress PGC1-α- and β-mediated mitochondrial biogenesis (Sahin *et al*, 2011; Chang *et al*, 2016). However, increased mitochondrial ROS driven by telomere dysfunction has been shown to induce DNA damage and activate a DDR in a positive feedback loop, making it experimentally complex to unravel which is the initiating step in the process (Passos *et al*, 2010).

There is still uncertainty regarding the physiological role of post-mitotic CM senescence during the ageing process. Previously, studies have attributed the beneficial effects of clearance of senescent cells in murine ageing heart to other proliferation-competent cells and not CMs (Baker *et al*, 2016). Studies have also proposed a role for telomere shortening in CM senescence: Short telomeres have been observed in aged murine CMs (Rota *et al*, 2007) and in cardiomyocytes from individuals with end-stage hypertrophic or dilated cardiomyopathy (Chang *et al*, 2018) and mouse models of accelerated telomere shortening exhibit cardiac dysfunction (Chang *et al*, 2016). However, whether short telomeres *per se* are causal in cardiomyocyte senescence during natural ageing has not been determined. Our data suggest that telomere length is not a limiting factor in CM senescence since (i) genetic elimination of p16$^{Ink4a}$-positive cells or treatment with senolytic drug, navitoclax, in aged mice did not affect telomere length or the frequency of very short telomeres and (ii) since super-resolution STED microscopy, which improved the resolution of clustered telomere signals, could not detect differences in FISH intensity between telomeres co-localising or not with γH2A.X. However, we acknowledge that there are limitations to our experimental approach: While interphase telomere FISH allows us to collect information regarding telomere length in specific cells of interest, it does not allow the recognition of telomeres in specific chromosomes or the detection of telomere-free ends. Furthermore, as we analyse 3-μm-thick tissue sections, we may not capture the entirety of the cardiomyocyte nucleus. Finally, while STED microscopy significantly improved resolution by threefold in XY and nearly 2.3-fold in Z compared to standard confocal microscopy, it is still possible that clusters of telomeres are not entirely resolved using this method.

Key questions remain regarding the consequences of CM senescence. While telomere dysfunction in CMs activates the classical senescence-inducing pathways, p21$^{CIP}$ and p16$^{Ink4a}$, CMs in contrast to cardiac stromal cells do not produce a classical SASP. Rather we have identified that factors such as Edn3, Tgfb2, and Gdf15 are released by senescent CMs and promote myofibroblast

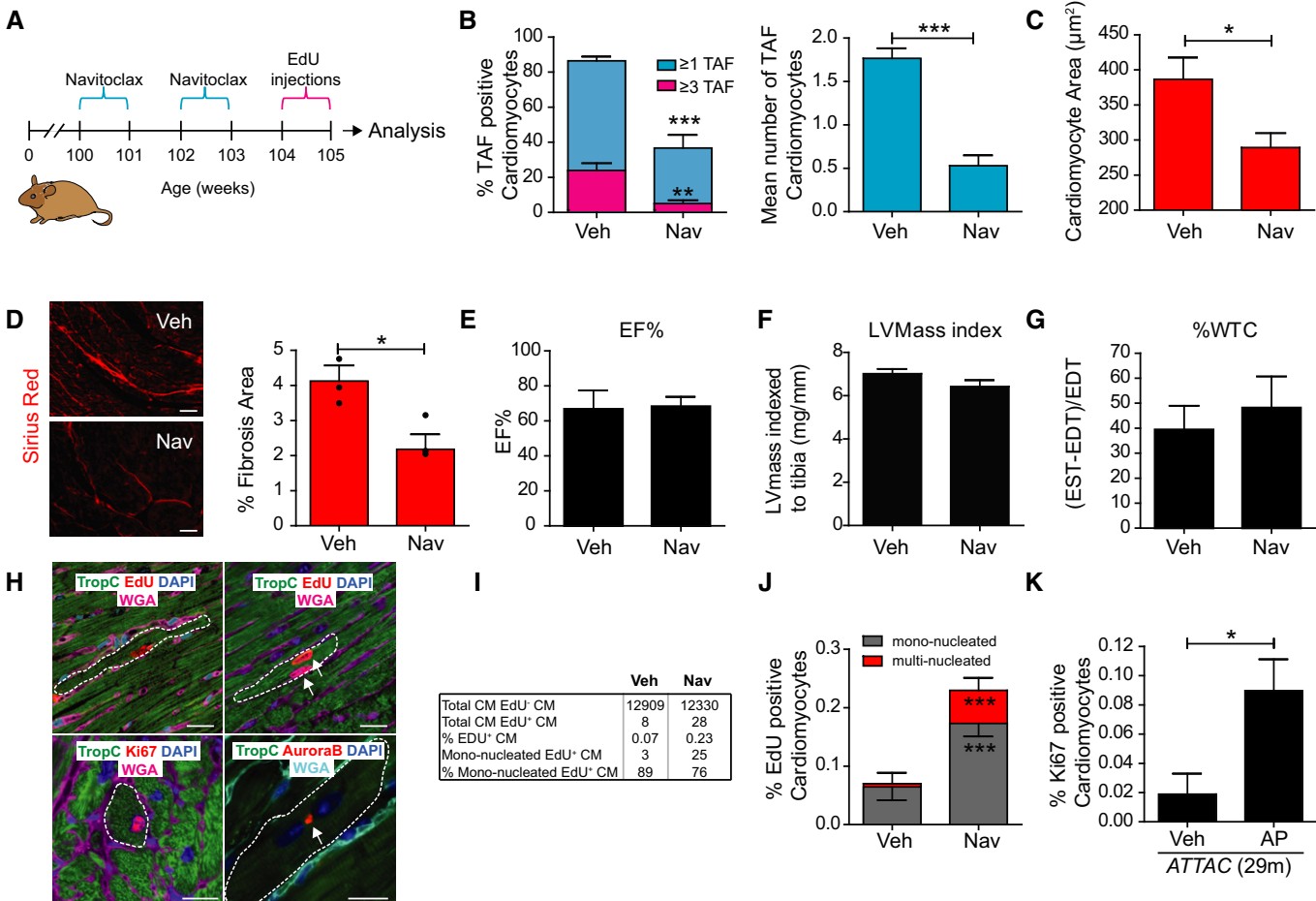

**Figure 8.  Pharmacological clearance of senescent cells with navitoclax (ABT 263) reduces cardiomyocyte senescence and stimulates cardiomyocyte regeneration.**

A   Scheme depicting experimental design. Mice at 100 weeks (23 months) of age were treated with vehicle (Veh) or navitoclax (Nav) intermittently for 2 weeks. At 104 weeks, mice were injected every day with EdU for 1 week.

B   Quantification of mean number of TAF and % of TAF-positive CMs in 24-month-old wild-type mice treated with vehicle or navitoclax (50 mg/kg/day). Data are mean ± SEM of *n* = 5–8 mice per group. More than 100 CMs were quantified per animal.

C   CM cross-sectional area in 24-month-old wild-type mice treated with vehicle or navitoclax. Data are mean ± SEM of *n* = 8 mice per group.

D   (Left) Representative images of Sirius red staining and (right) % of fibrosis area in 24-month-old mice treated or not with navitoclax. Data are mean ± SEM of *n* = 3 per group. Scale bar represents 50 μm.

E–G  MRI analysis of ejection fraction (EF%), left ventricle mass (LVmass) index and % of ventricle wall thickness (%WTC) in 24-month-old mice treated or not with navitoclax. Data are mean ± SEM of *n* = 6 mice per treatment group.

H   Examples of confocal microscopy images of CMs positive for CM marker troponin-C (TropC), EdU, Ki-67, and Aurora B from navitoclax-treated animals. In the upper right panel, white arrows identify two nuclei in the same CM that have incorporated EdU. In the lower right panel, white arrows identify EdU-expressing Aurora B symmetrically between two nuclei. Scale bars represent 20 μm.

I   Table summarising numbers of CMs quantified in (J).

J   Quantification of EdU-positive CMs (mono- or multi-nucleated) in vehicle- and navitoclax-treated animals. Data are mean ± SEM of *n* = 5–6 mice per group.

K   % of Ki-67-positive CMs in 29-month-old INK-ATTAC mice treated with vehicle or AP20187. Data are mean ± SEM of *n* = 4–6 mice per group.

Data information: Asterisks denote a statistical significance using two-tailed *t*-test (B, C, K) or Mann–Whitney test (D, J). ***P < 0.001; **P < 0.01; *P < 0.05.

activation and CM hypertrophy. Based on these data and the observed reduction in myocardial remodelling following senescent-cell elimination, we suggest that CM senescence and the SASP contribute to age-related cardiac dysfunction clinically. As such, they represent therapeutic targets to improve myocardial health in the older population.

Another important question, which arises from this study, is whether loss of senescent cardiomyocytes by senotherapies may

be detrimental for the heart, which may limit its clinical potential. Our study, in accordance with others (Baker *et al*, 2016; Roos *et al*, 2016; Demaria *et al*, 2017), shows that the loss of senescent cardiomyocytes does not adversely alter cardiac function in aged mice. Based on our observations, we would suggest that function is maintained as a result of compensatory mechanisms that include CM renewal and replacement, as indicated by the increased frequency of EdU- and Ki-67-positive mononuclear CMs

and the observation of CMs expressing Aurora B symmetrically between nuclei (Hesse *et al*, 2018). Following senescence clearance, we also identified an increase in CM karyokinesis. A process that can be associated the CM hypertrophy that occurs during adaptive and ultimately maladaptive remodelling in response to pathological conditions which involve CM loss, including myocardial infarction and pressure overload (Ahuja *et al*, 2007). While the observed global reduction in hypertrophy argues against karyokinesis and hypertrophy being the main compensatory mechanisms following senescent CM elimination, extended longitudinal studies of cardiac function will need to be conducted to ascertain whether maladaptive remodelling and cardiac dysfunction occur in the longer term.

In summary, our study provides evidence for the concept that post-mitotic CM senescence is a major effector of myocardial ageing and offers a proof-of-principle that modulation of cardiac senescence is a viable treatment strategy. While described in myocytes, we speculate that our proposed mechanism may explain PoMiCS which has been detected in other tissues such as neurones (Jurk *et al*, 2012), osteocytes (Farr *et al*, 2017), and adipocytes (Minamino *et al*, 2009).

# Materials and Methods

### Animals and procedures

*Ageing studies*
Mixed-sex C57BL/6 mice were analysed at either 3, 15, 20, 24 or 30 months of age. For whole-body X-ray irradiation, mixed-sex 1-month-old C57BL/6 mice were irradiated with 2 Gy followed by 11-month recovery period before culling. *TERC$^{-/-}$ C57BL/6 mice.* Male mice were bred to produce successive generations of mice with decreasing telomere length. Hearts from fourth-generation (G4) mixed-sex mice were studied. *Catalase$^{-/-}$ and MnSOD$^{+/-}$ mice.* Mouse models for elevated reactive oxygen species and mitochondrial dysfunction: Catalase$^{-/-}$ and MnSOD$^{+/-}$ were aged until 22–24 months. *PolgA$^{mut/mut}$ mice.* Mice with a knock-in missense mutation (D257A) at the second endonuclease-proofreading domain of the catalytic subunit of the mitochondrial DNA (mtDNA) polymerase Polγ (*PolgA$^{mut/mut}$* mice) and *PolgA$^{+/+}$* mice were used at 12 months of age. Mice were group-housed in individually ventilated cages with a constant temperature of 25°C, a 12-h light/dark cycle and with RM3 expanded chow (Special Diet Services).

For the above mice projects were approved Newcastle University Animal Welfare Ethical Review Board and experiments were conducted in compliance with the UK Home Office (PPL P3FC7C606 or 60/3864).

*MAO-A transgenic mice*
MAO-A mice, on the C57BL6/J background, with cardiac-specific overexpression of MAO-A driven by the α-MHC promoter are previously described (Villeneuve *et al*, 2013). Mixed-sex MAO-A offspring and non-transgenic littermates were used for the experiments at 6 months of age. N-acetyl-cysteine was provided at 1.5 g/kg/day between 1 and 6 months of age. Mice were housed in a pathogen-free facility (B 31 555 010). Experiments were approved by University of Toulouse local ethic committee (CEEA-122

2015-01) and conformed to the Guide for the Care and Use of Laboratory Animals published by the Directive 2010/63/EU of the European Parliament.

*INK-ATTAC mouse model for the clearance of senescent cells*
Experimental strategy for making transgenic mice with a senescence-activated promoter coupled to the drug-activatable ATTAC "suicide" transgene and GFP was devised by JLK and TT. INK-ATTAC mice were created and characterised at Mayo through a collaboration among the JLK-TT, NKL and van Deursen laboratories. Animals were crossed onto a C57BL/6 background (van Deursen laboratory) and bred, genotyped and aged (JLK-TT laboratory). Mixed-sex mice were housed in a pathogen-free facility, at 2–5 mice/cage with 12-h light/12-h dark cycle at 24°C and *ad libitum* access to standard mouse diet (Lab Diet 5053, St Louis, MO, USA) and water. AP20187 (10 mg/kg) or vehicle was administered to 27-month-old mice by intraperitoneal injection every 3 days for 2 months.

*Senolytic treatment of male C57BL/6 mice*
Mice at 22 months of age were purchased from Charles River (Charles River Laboratories International, UK). Mice were randomly assigned to a treatment group. ABT263 (navitoclax) or vehicle alone was administered to mice by oral gavage at 50 mg/kg body weight per day (mg/kg/day) for 7 days per cycle for two cycles with a 1-week interval between the cycles. Hearts were collected directly into 50 mM KCl to arrest in diastole.

For all animal studies, details of sample sizes are included in the figure legends.

*Human tissue collection and ethics*
Human heart tissue was obtained from male and female patients undergoing open-heart surgery for aortic stenosis, with a section of the right atrial appendage being placed in 10% neutral buffered formalin (VWR, 9713.9010) immediately after dissection. Subsequent processing steps for paraffin embedding were the same as for mouse tissue (as described below). All tissue samples were obtained under the clause in the Human Tissue Act that enables anonymised samples to be taken without consent in the context of an ethically approved study. This study was approved by the Research Ethics Committee, UK, REC reference number: 10/H0908/56.

*Adult mouse cardiomyocyte isolation and purification*
Hearts from mixed-sex CJ57/BL6 mice were placed on a Langendorff setup. Cell suspensions were obtained by enzymatic digestion and sedimentation. To enrich the CM population, cells were stained at 4°C for 30 min with a cocktail of biotin-conjugated antibodies (CD31 (390), CD45 (30-F11), and Sca-1 (D7); BioLegend) and the EasySep™ Mouse Biotin Positive Selection Kit, used to remove the labelled non-CM cells. Cultured CM purity was assessed by immunofluorescent staining against CD31, Sca-1, CD45, and α-actinin. Qiagen RNA extraction kit was used of RNA isolation. CMs were cultured on laminin-coated wells in MEM with Hank's salts and L-glutamine (SIGMA), 0.1 mg/ml BSA, 10 mM butanedione monoxime, ITS Liquid Media Supplement (SIGMA).

*Culture of H9C2 rat myoblasts*
H9C2 rat heart-derived embryonic myocytes (ATCC) were cultured in DMEM (SIGMA), 5% foetal calf serum, penicillin/streptomycin

100 U/ml, 2 mM glutamine, at atmospheric conditions (air plus 5% $CO_2$).

### Neonatal rat ventricular myocyte (NRVM) isolation

For neonatal cardiomyocytes, hearts of mixed-sex 2- to 3-day-old Sprague Dawley rats were dissociated with 0.43 mg/ml collagenase type A (Roche) and 0.5 mg/ml pancreatin (Sigma). Myocyte enrichment was performed by centrifugation through a discontinuous Percoll gradient, and the resultant suspension of myocytes was plated onto gelatin-coated culture dishes in mix medium containing 80% DMEM High Glucose and 20% M199, supplemented with 10% FBS, 5% HS and 1% antibiotics. After plating, cells were cultivated in mix medium with reduced FBS concentration (5%). Plasmid transfections were performed using Lipofectamine 2000 reagent (Life Technologies).

### Mouse embryonic cardiomyocyte isolation and culture

Under sterile conditions, hearts were removed from mixed-sex E17.5 embryos and then cut into multiple fragments in cardiomyocyte balanced salt buffer (CBSB; 116 mM NaCl, 20 mM HEPES, 1 mM $NaH_2PO_4$, 5.365 mM KCl, 831 nM $MgSO_4$). Heart fragments were then incubated in CBSB with 80 U/ml collagenase type II (Worthington) for 5 min. Allowing fragments to settle, supernatant was removed and centrifuged at $700 \times g$ for 5 min. Remaining fragments were then re-suspended in cardiomyocyte enzyme solution (CBSB, 80 U/ml collagenase II and 0.25 mg/ml trypsin) for 30 min, with gentle shaking. Fragments were allowed to settle, and then, supernatant containing dissociated cells was removed and centrifuged at $700 \times g$ for 5 min, washed in FBS and stored at 4°C. Remaining fragments were then re-suspended as before into cardiomyocyte enzyme solution. After centrifugation, the supernatant was discarded and pellet was again re-suspended in 4°C FBS and placed on ice. Re-suspension, centrifugation and collection were then repeated until all fragments had been dissolved (7–10 cycles). All FBS suspensions were pooled together and centrifuged at $700 \times g$ for 5 min. The supernatant was then discarded and cells re-suspended in cardiomyocyte growth medium (Dulbecco's Modified Eagle's Medium (DMEM), supplemented with 17% Medium 199, 5% FBS, 10% horse serum (SIGMA, H0146), 100 µg/ml streptomycin, 100 units/ml penicillin and 2 mM L-glutamine). Cells were then seeded into a collagen-coated (1 mg/ml) T75 culture flask (SIGMA) and incubated at 37°C for 2 h. After the incubation, the supernatant, containing cardiomyocytes, was collected from the flask and the adherent fibroblasts discarded. Cardiomyocytes were then cultured in Cardiomyocyte Growth Medium.

### Culture and differentiation of AC10 cells

Cells were seeded at a density of $2.5 \times 10^4$/ml on glass coverslips coated with 12.5 µg/ml fibronectin in 0.02% gelatin in DMEM/F-12 with 12.5% FBS. Once confluent, medium was then replaced by mitogen-deficient medium, DMEM/F-12 supplemented with 2% HS and insulin–selenium–transferrin supplement (ITS, Invitrogen), and the cells cultured for a further 2 weeks to allow terminal differentiation.

## Cell culture, transfections and treatments

### Transfection of H9C2 myoblasts and rat neonatal cardiomyocytes

Cells were transiently transfected with a 53BP1-GFP reporter protein using the plasmid pG-AcGFP-53BP1c or either TRF1-FokI-D450A or TRF1-FokI or pBABe-HA-ER-IPpoI plasmids using Lipofectamine 2000 (ThermoFisher) at a ratio of 3 µl Lipofectamine 2000 to 1 µg DNA following the manufacturer's protocol. Transfected cells detected with an anti-FLAG antibody (SIGMA).

### Stress-induced senescence of cultured cells

For H9C2 cells, rat neonatal or mouse embryonic CM senescence was induced by 10 Gy X-ray irradiation. In both cell types, we observed senescence after 10 days using SA-β-Gal assay (> 70% positive cells) and absence of proliferation marker Ki-67. H9C2 senescent or proliferating cells were treated with ABT-263 (Navitoclax; AdooQ Bioscience, USA), and viability was assessed using a Tali Image-Based Cytometer (Invitrogen).

### EdU incorporation assays

Cells were incubated in 10 µM EdU in normal growth medium for 24 h. EdU was detected using the Click-iT® EdU Imaging Kit (Invitrogen), as per manufacturer's protocol.

### Thoracic Irradiation

A TrueBeam linear accelerator (Varian Medical Systems, Palo Alto) was used for the mouse studies. Anesthetised INK-ATTAC mice (2 months of age) received a single, 20 Gy radiation dose delivered to the thoracic region. The radiation beam was collimated to an area encompassing the mouse lungs and the radiation field positions on the mice were verified using kV-CBCT and 2D kV imaging of the animals prior to irradiation. The dose rate at the prescription point was 14.8 Gy/min, using 89 cm source to the surface distance. Dose was prescribed to midline in the mice and was confirmed using film and ion chamber dosimetry. All mice prophylactically received 100 mg/ml Baytril antibiotic in their drinking water for 3 weeks post-irradiation. One month following irradiation, mice were randomised to AP20187 (10 mg/kg), delivered by intraperitoneal injection (treatments twice weekly) or vehicle groups. Body weight was monitored weekly. Mice were euthanised 6 months post-irradiation exposure using a lethal dose of pentobarbital. Animal experiments involving INK-ATTAC mice were performed under protocols approved by the Mayo Clinic Institutional Animal Care and Use Committee.

### Live-cell imaging

For live-cell time-lapse microscopy of 53BP1 reporter fluorescence, H9C2 cells were plated on glass coverslip-bottomed dishes (MatTek), and cells were imaged every 10 min for 10 h as Z-stacks over 7 µm with a 63× objective (NA = 1.4), using a Zeiss spinning disc confocal microscope with cells incubated at 37°C with humidified 5% $CO_2$. AcGFP-53BP1 foci dynamics were analysed using Imaris Software.

### Immunohistochemistry

Deparaffinisation, hydration and antigen retrieval of formalin-fixed paraffin-embedded heart tissues were performed as previously described. Sections were incubated with M.O.M mouse IgG blocking reagent (90 µl blocking reagent: 2.5 ml TTBS) for 1 h at room temperature. After 2 × 5 minute PBS washes, sections were incubated with avidin for 15 min, rinsed with PBS, and then incubated with biotin for 15 min at room temperature. Primary antibody was then diluted in M.O.M diluents (7,500 ml TTBS: 600 µl protein

concentrate) and incubated overnight at 4°C. After 2 × 5 min PBS washes, sections were incubated in biotinylated-mouse IgG reagent diluted in blocking solution (1:200) for 30 min at room temperature. After 2 × 5 min PBS washes, endogenous peroxidase activity was blocked by incubating sections in 0.9% $H_2O_2$ in water for 30 min. After 2 × 5 min PBS washes, sections were incubated in AB complex for 30 min at room temperature. After 3 × 5 min PBS washes, sections were then incubated in NovaRed solution for up to 10 min, rinsed with water, counterstained with haematoxylin for 2 min, washed in PBS and then transferred to ammonia water for 30 s. After a 1-min wash in water, sections were then dehydrated through 70, 90 and 100% ethanol and then histoclear for 5 min each. Sections were then mounted with DPX.

Primary antibodies used were as follows: mouse monoclonal anti-4HNE (1:100, MHN; JaICA), mouse monoclonal anti-8-OHdG (1:100, MOG; JaICA) and rabbit polyclonal anti-p21 (1:200, ab7960; Abcam).

### Immunofluorescence

Cells were grown on sterile coverslips, fixed with 2% PFA and permeabilised with PBG-T (PBS, 0.4% fish-skin gelatin, 0.5% BSA, 0.5% Triton X-100). Cells were incubated in primary antibody for 1 h at room temperature (RT), followed by incubation in secondary antibody for 45 min. Primary antibodies: anti-γ-H2AX (1:200, #9718; Cell Signaling), anti-53BP1 (1:200, #4937; Cell Signaling), anti-53bp1 (1:200, nb100-305, Novus Biologicals), anti-α-actinin (1:100, A7811; SIGMA), anti-troponin I (1:200, Catalogue, Company), anti-troponin-C (1:200, Abcam ab30807), anti-PCM1 (1:200, ab154142; Abcam), anti-FLAG (1:1,000, F1804; SIGMA) and anti-Ki-67 (1:250 (4 μg/ml), ab15580; Abcam). Secondary antibodies: Alexa Fluor® 488, Alexa Fluor® 594 or Alexa Fluor® 647 (Invitrogen).

### Microscopy

For fluorescence microscopy, a Leica DM5500B wide-field fluorescence microscope and Zeiss AxioObserver Spinning Disk confocal microscope were used. For super-resolution STED microscopy, a Leica SP8 confocal (inverted) gSTED 3D super-resolution microscope was used. For all quantitative immunohistological analysis studies were performed in a blinded manner.

### Immuno-FISH and FISH

Immuno-FISH was performed as described in Hewitt *et al* (2012). Briefly, cells were grown on coverslips and immunocytochemistry performed as described above, with either anti-γH2AX (1:200, 9718; Cell Signaling) or anti-53BP1 (1:200, 4937; Cell Signaling). Cells were then fixed with methanol: acetic acid (3:1), dehydrated through an alcohol series, air-dried and incubated with PNA hybridisation mix [70% deionised formamide (SIGMA), 25 mM $MgCl_2$, 1 M Tris pH 7.2, 5% blocking reagent (Roche) containing 2.5 μg/ml Cy3-labelled telomere-specific (CCCTAA) peptide nucleic acid probe (Panagene)], for 10 min at 80°C in a then for 2 h at RT in the dark humidifier chamber. Cells were washed with 70% formamide in 2× SSC for 10 min, 2× SSC for 10 min and PBS for 10 min. Cells were mounted with ProLong® Gold Antifade Mountant with DAPI (ThermoFisher). For immuno-FISH in formalin-fixed paraffin-embedded tissues, sections were de-waxed and rehydrated through an alcohol series. Antigen retrieval was via 0.1 M citrate buffer boiling. Sections were

incubated in blocked buffer (1:60 Normal Goat Serum in PBS/0.1% BSA) followed by primary antibody at 4°C. Sections were then incubated with a biotinylated secondary antibody (1:200) followed by avidin-DCS (1:500). Sections were post-fixed in 4% PFA for 20 min, and FISH was carried out as described above. For centromere-FISH (SADS detection), a FAM-labelled, CENPB-specific (centromere; ATTCGTTGGAAACGGGA) peptide nucleic acid probe (Panagene) was used. Frequency of SADS was assessed by quantification of decondensed/elongated centromeres.

### RNA *in situ hybridisation*

RNA-*ISH* was performed after RNAscope protocol from Advanced Cell Diagnostics, Inc. (ACD). Paraffin sections were deparaffinised with Histoclear, rehydrated in graded ethanol (EtOH), and $H_2O_2$ was applied for 10 min at RT followed by two washes in $H_2O$. Sections were placed in hot retrieval reagent and heated for 30 min. After washes in $H_2O$ and 100% EtOH, sections were air-dried. Sections were treated with protease plus for 30 min at 40°C, washed with $H_2O$ and incubated with target probe (p16, eGFP) for 2 h at 40°C. Afterwards, slides were washed with $H_2O$ followed by incubation with AMP1 (30 min at 40°C) and next washed with wash buffer (WB) and AMP2 (15 min at 40°C), WB and AMP3 (30 min at 40°C), WB and AMP4 (15 min at 40°C), WB and AMP5 (30 min at RT) and WB and, finally, AMP6 (15 min at RT). Finally, RNAscope 2.5 HD Reagent kit-RED was used for chromogenic labelling. After counterstaining with haematoxylin, sections were mounted and coverslipped. All analysis was performed in a blinded manner.

### Crosslinked chromatin immunoprecipitation assay (ChIP)

Three-month-old and 30-month-old frozen heart tissues ($n = 3$ per age group) were powdered using dry pulverisation cryoPREP (Covaris). The powder was then re-suspended into 1% formaldehyde solution in PBS and cells cross-linked for 5 min at RT, then quenched with glycine buffer at final concentration of 125 mM. The cells were rinsed in cold PBS and harvested into PBS with protease inhibitors and centrifuged for 5 min, 4°C, 1,000 × g. The pellet was re-suspended in ChIP lysis buffer (1% SDS, 10 mM EDTA, 50 mM Tris–HCl pH 8.1) and incubated for 20 min on ice. Lysate was sonicated to generate average size fragment of 200–700 bp; cell debris was pelleted by centrifugation for 10 min, 4°C, 8,000 × g. The supernatant contained chromatin, which was quantified using $OD_{260}$ on Nanodrop.

Twenty-five micrograms of chromatin per IP was diluted 1:10 with dilution buffer (1.1% Triton, 1.2 mM EDTA, 16.7 mM Tris–HCl pH 8.1, 167 mM NaCl, with protease inhibitors) and precleared by incubating with 25 μl blocked Staph A membranes (Isorbin) for 15 min. Precleared chromatin was then incubated with 5 μg anti-γH2AX (phospho S139; Abcam, ab2893) and species- and isotype-matched control ChIP grade IgG (Abcam, ab46540) overnight at 4°C. Immunoprecipitated chromatin was collected with blocked Staph A membranes, which were then sequentially washed with cold buffers; Low Salt Wash Buffer (0.1% SDS, 1% Triton X-100, 2 mM EDTA, 20 mM Tris–HCl pH 8.0, 150 mM NaCl), High Salt Wash Buffer (0.1% SDS, 1% Triton X-100, 2 mM EDTA, 20 mM Tris–HCl pH 8.0, 500 mM NaCl), LiCl Wash Buffer (0.25 M LiCl, 1% NP-40, 1% sodium deoxycholate, 1 mM EDTA, 10 mM Tris–HCl pH 8.0) and two washes in TE Buffer (10 mM Tris pH 8.0, 1 mM EDTA). The immunoprecipitated

chromatin was eluted in 500 μl Elution buffer (1% SDS, 100 mM NaHCO$_3$), cross-links reversed and DNA obtained by phenol:chloroform extraction and gel purification. Real-time PCR specific for telomeric repeats was performed as described in Hewitt *et al* (2012). Briefly, 4 μl of ChIP eluate (bound fraction) was added to a final volume of 13 μl quantitative PCR containing 6.5 μl Jumpstart SYBR master mix (Sigma) and 10 pmol primer mix. Telomere PCR conditions were as follows: 10 s at 95°C followed by 40 cycles of 10 s at 95°C, 30 s at 55°C, and 30 s at 72°C followed by ABI7500 Fast Real-Time PCR System machine predetermined melt curve analysis (Applied Biosystems). Total input fraction was also collected from chromatin samples. Bound to input (B/I) ratios were determined for each amplicon by taking a 4 μl aliquot of the DNA extracted from the input and bound samples. These were amplified together with a defined amount of genomic DNA on the same plate, using the latter to construct the standard curve. Isotype-matched irrelevant antibody control (IgG) value was subtracted from the result before plotting the graph. Each PCR was performed in triplicate and results calculated using the following equation: $(1/(2A)) \times 100$.

### Senescence-associated β-galactosidase activity assay

SA-β-Gal activity assay was performed as described previously (Dimri *et al*, 1995). For *in vitro* studies, cells were grown on sterile coverslips, and for *in vivo* studies, samples were cryo-embedded and staining performed on 10 μm sections. Samples were fixed with 2% paraformaldehyde in PBS for 5 min, washed with PBS for 5 min and then incubated at 37°C overnight in SA-β-Gal solution containing 150 mM sodium chloride, 2 mM magnesium chloride, 40 mM citric acid, 12 mM sodium phosphate dibasic, 5 mM potassium ferricyanide, 5 mM potassium ferrocyanide and 1 mg/ml 5-bromo-4-chloro-3-indolyl-β-d-galactosidase (X-Gal) at pH 5.5. For *in vivo* staining and quantifications, sections were co-labelled with anti-troponin-C and WGA as above. All analysis was performed in a blinded manner.

### TRAP assay

Heart samples were snap-frozen in liquid nitrogen immediately after dissection. Using a liquid nitrogen-cooled pestle and mortar, tissues were ground into a fine powder. Telomerase activity was then determined following the TeloTAGGG Telomerase PCR ELISA kit (Roche).

### Assessment of cardiomyocyte hypertrophy

Hypertrophy was assessed by cross-sectional area using established methods as described (Shenje *et al*, 2014). Troponin-C was used to identify CMs and membranes labelled with wheat germ agglutinin (WGA; Alexa Fluor® 647, W32466, Invitrogen). Only CMs in the left ventricle free wall were analysed. To control for tissue orientation only myocytes that were surrounded by capillaries, all displaying a cross-sectional orientation were analysed. All analysis was performed in a blinded manner.

### Evaluation of cardiomyogenesis in mice

To allow retrospective quantification of cumulative proliferation and binucleation, occurring the week following navitoclax treatment, mice were injected (intraperitoneal) with EdU (100 μg/g body weight) once daily every 24 h for seven consecutive days (Fig 7A). More than five animals were studied for each treatment group. EdU

incorporation was detected using the Click-iT™ EdU Imaging Kit (ThermoFisher). Sections were also labelled with primary antibody for troponin-C and the membrane-specific label WGA. A Leica SP5 laser scanning confocal system was used for the analysis. Analysis was carried out in a blinded fashion similar to our previous studies (Richardson *et al*, 2015). In brief, myocardial tissue was sectioned transversely in 40 μM sections, with sections separated by intervals of 200 μM, allowing representative analysis throughout the ventricle. Over 80 images were taken, in a random fashion, spanning over eight individual sections per heart. Analysis included the assessment of EdU labelling in > 1,500 cardiomyocytes per heart. Incorporation of EdU in the cardiomyocyte population was quantified based on the detection of EdU labelling in the nucleus of a troponin-C-expressing cell. To discriminate proliferation from binucleation (karyokinesis without cytokinesis), the number of EdU-labelled nuclei within each Edu$^+$ cardiomyocyte was assessed using WGA to identify the cell membrane. The incorporation of EdU into a cell with only one nucleus was considered to be a result of proliferation, when two or more EdU-labelled nuclei were found in the same cardiomyocyte, this was considered to have been incorporated during a multi-nucleation event. Ki-67 or Aurora B-expressing cardiomyocytes were identified as co-expression of Ki-67/Aurora B and troponin-C within a single-cell membrane (based on WGA labelling). The total percentage of positively expressing cardiomyocytes was calculated as previously described (Richardson *et al*, 2015), based on the analysis of > 4,000 cardiomyocytes per heart. All quantifications were performed blinded using digital image analysis (ImageJ; U.S. National Institutes of Health; http://rsbweb.nih.gov/ij/).

### MR imaging and analysis

Magnetic resonance images were acquired with a horizontal bore 7.0T Varian microimaging system (Varian Inc., Palo Alto, CA, USA) equipped with a 12-cm microimaging gradient insert (40 gauss/cm) and analysed as described previously (Redgrave *et al*, 2017). Percentage change in wall thickening was calculated from the mean of wall thickness at four points in all slices at end systole and end diastole thickness. % change = $(EST - EDT)/EDT \times 100$. All analysis was performed in a blinded manner.

### RNA sequencing

Paired-end reads were aligned to the mouse genome (mm9) using a splicing-aware aligner (tophat2). Only unique reads were retained. Reference splice junctions were provided by a reference transcriptome (Ensembl build 67), and novel splicing junctions determined by detecting reads that spanned exons that were not in the reference annotation. True read abundance at each transcript isoform was assessed using HTSeq (Python) before determining differential expression with the tool DESeq2, which models mean-variance dependence within the sample set. Significance was determined using an FDR-corrected *P*-value ≤ 0.05.

### Computational modelling

Two different models were constructed to examine different hypotheses on how TAF accumulate in cells. The models were encoded in the Systems Biology Markup Language (SBML) modelling standard (Hucka *et al*, 2003) with the Python SBML shorthand tool (Wilkinson, 2011). Model simulations were carried out in COPASI (Hoops *et al*, 2006), and the results were analysed and plotted in R

using ggplot2 (Wickham, 2009). We used stochastic simulation (Gillespie direct method) in order to be able to account for the variability in the experimental data. The models were deposited in BioModels (Chelliah *et al*, 2015) and assigned the identifiers MODEL1608250000 and MODEL1608250001.

*Cufflinks FPKM determination*
To calculate library normalised read counts for each transcript in every replicate, we used the tool Cufflinks, which returns the proportion of reads per million, which mapped to each gene isoform.

*Gene set enrichment analysis (GSEA)*
We perform a GSEA by creating a ranked list of the DESeq2 results file, where genes are ranked by their log-transformed, non-FDR-corrected *P*-values. This list is then used as an input to the GSEA software, which assesses positive or negative shifts of GO ontology classes in the global distribution of gene expression. This is achieved by the calculation of an enrichment score, which reflects the movement of a particular ontological class to the positive and negative extremes of the distribution of the expression of all genes. Permutation testing then allows for the estimation of significance prior to FDR correction.

*Heatmaps*
Heatmaps were created in R using the *ggplots* package. These heatmaps displaying normalised (row scaled—Z-score) FPKM gene expression across a series of replicates were clustered using a Pearson correlative clustering approach in the "hclust" R package.

*Principal component analysis*
Principal component analysis was performed in R using the *prcomp* method.

**Statistical analysis**

We conducted two-tailed *t*-tests, one-way ANOVA, Gehan–Breslow and Mann–Whitney tests using GraphPad Prism.

# Data accessibility

The RNA-seq data from this publication have been submitted to the GEO database (http://www.ncbi.nlm.nih.gov/geo/), accession number: GSE95822. The models were deposited in BioModels and assigned the identifiers MODEL1608250000 and MODEL1608250001 (https://www.ebi.ac.uk/biomodels-main/).

**Expanded View** for this article is available online.

## Acknowledgements
This work was funded by BBSRC grants BB/H022384/1 and BB/K017314/1 to JFP, BHF project grant PG/15/85/31744 to GR and JFP, NIH grant AG013925 and funding from the Connor Group and Noaber Foundation to JLK, and grant from Région Occitanie, France and a grant from foundation Cariplo 2014-0672, to JMP, DM and VDE. CJP and JFP were funded by the MRC-Arthritis Research UK Centre for Integrated research into Musculoskeletal Ageing (CIMA) grant MR/K0063121/1. We thank Prof. Roger Greenberg for kindly providing us with the TRF1-FOKI plasmid. We thank Gabriele Saretzki for expert assistance in TRAP assays, Thomas von Zglinicki for providing mice tissues and Glyn Nelson for assistance in live-cell microscopy. We thank Gaelle Legube for technical advice regarding the use of the I-PpoI endonuclease.

## Author contributions
RA, DM and AL performed the majority of experiments. JC, JB, HS, MO, DJ, CP, CC-M, ED, AW, EF, RB-P, SV, ST-C, KLK, VD-E, MJS, CMR, NR, JM, TT performed and evaluated individual experiments; AO, LG, HMA, AP, NKLB, JM, JLK and PDA designed and supervised individual experiments; LG provided materials; JM-P, GDR and JFP designed and supervised the study; JFP, RA and GDR wrote the manuscript with contributions from all the authors.

## Conflict of interest
Patents on INK-ATTAC mice are held by Mayo Clinic and licensed to Unity Biotechnology. J.L.K. and T.T. may gain financially from these patents and licences. This research has been reviewed by the Mayo Clinic Conflict of Interest Review Board and was conducted in compliance with Mayo Clinic Conflict of Interest policies. The remaining authors declare no competing financial interests.

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
