## [Review Process File · The EMBO Journal]

Length-independent telomere damage drives Post-mitotic cardiomyocyte senescence

Rhys Anderson, Anthony Lagnado, Damien Maggiorani, Anna Walaszczyk, Emily Dookun, James Chapman, Jodie Birch, Hanna Salmonowicz, Mikolaj Ogrodnik, Diana Jurk, Carole Proctor, Clara Correia-Melo, Stella Victorelli, Edward Fielder, Rolando Berlinguer-Palmini, Andrew Owens, Laura Greaves, Kathy L. Kolsky, Angelo Parini, Victorine Douin-Echinard, Nathan K. LeBrasseur, Helen M. Arthur, Simon Tual-Chalot, Marissa J. Schafer, Carolyn M Roos, Jordan Miller, Neil Robertson, Jelena Mann, Peter D. Adams, Tamara Tchkonina, James L Kirkland, Jeanne Mialet-Perez, Gavin D Richardson & João F. Passos

Review timeline:

Submission date:	16th Aug 2018
Editorial Decision:	21st Sep 2018
Revision received:	27th Nov 2018
Editorial Decision:	14th Dec 2018
Revision received:	18th Dec 2018
Accepted:	2nd Jan 2019

Editor: Daniel Klimmeck

Transaction Report:

1st Editorial Decision

21st Sep 2018

Thank you for the submission of your manuscript (EMBOJ-2018-100492) to The EMBO Journal. Your manuscript has been sent to three referees, and we have received reports from all of them, which I enclose below.

As you will see, the referees acknowledge the potential high interest and novelty of your work, although they also express a number of issues that will have to be addressed before they can support publication of your manuscript in The EMBO Journal. While referees #1 and #2 are overall more positive, referee #3 states that the conceptual advance provided is not sufficient in his/her view (ref#3, pt.1). In addition, referee #3 questions the physiological relevance of your results and points to inconsistencies in the data (ref#3, pts.2,3). In addition, the referees state issues related to literature references, experimental design, documentation of methodologies as well as missing controls that would need to be conclusively addressed to achieve the level of robustness needed for The EMBO Journal.

I judge the comments of the referees to be generally reasonable and given their overall interest, we are in principle happy to invite you to revise your manuscript experimentally to address the referees' comments. I agree that strengthening the physiological implication of your results would be important to achieve a coherent study.

REFEREE REPORTS:

Referee #1:

This is an interesting manuscript describing the accumulation of persistent DNA damage at telomeres in cardiomyocytes and improved health upon removal of senescent cells.

I have relatively few suggestions to improve this manuscript:

- Telomere clustering has not been fully ruled out in my opinion. Measuring the number of telomeric signals per cell may give an indication that indeed the techniques employed really allow the detection of all 92 individual telomeres expected in a nucleus thus excluding clustering that could reduce their apparent number and artefactually increase their length.
- Fig S1b needs to be complemented by a quantification of 53bp1 total number of foci as in 1d
- ChIP in 1f need a negative control/input normalizer: Alu PCR?
- Irreparability of telomeres was demonstrated at the molecular level by Fumigalli et al and it would be fair to quote this along Hewitt et al wherever this report is duly quoted
- Experiments with TRF1-Fok1 are interesting but lack an adequate control: a nuclease that induces a similar number of DSB in not telomeric regions. That is essential to support the claims made.
- I would recommend complementing the results obtained with conditioned medium from old CM by testing its impact on DDR and senescence markers: are they induced?
- It may be appropriate to quote this recent publication
<https://www.ncbi.nlm.nih.gov/pubmed/30150400>

Referee #2:

Reductions in telomere length have been implicated in cellular senescence. The authors address the question how ageing affects cellular senescence in postmitotic cardiac myocytes, which have a very low proliferative activity and thus telomere shortening is unlikely to be a trigger in these cells. An impressive amount of experimental data using ageing mice and an additional 6 KO and transgenic lines plus primary cultures, cardiac cell lines and a total of 8 primary panels and 12 supplements, all packed with data lead to very convincing conclusion that mitochondria derived reactive oxygen species probably trigger DNA damage leading to the binding of proteins of the DNA damage response such as γ H2A.X, 53BP1 to the telomere triggering a senescent program the animals demonstrated an age-dependent increase in expression of CDK inhibitors p16INK4a, p21CIP, and p15 INK4a in cardiac myocytes. Senescence in proliferating cells is also associated with a specific secretory pathway, which however seems to be absent in cardiac myocytes. Nonetheless some cytokines such as endothelin 3, TGF β 2 and GDF15 were found to be upregulated in old hearts. Using a number of genetic and experimental approaches the authors show that the trigger of age - dependent cardiac myocyte senescence appears to be enhanced mitochondrial ROS activity. Senescent hearts show an increase in cardiac hypertrophy, wall rigidity and fibrosis while ejection fraction is preserved. Genetic and pharmacological approaches were developed to remove specifically senescent cells from the ageing heart, which normalize cardiac fibrosis and cardiac hypertrophy. Some of the data suggest that the clearance of senescent myocytes may trigger activation of resident stem cells to replenish the cardiac myocyte pool with young myocytes as total cell number and the size of the ventricular wall remain unchanged.

The experiments have been executed to a very high standard and apart from a quantification of the reduction in fibrosis in the genetic approach to remove senescent cells (INK-ATTAC), I have no specific suggestions for additional experiments.

Minor

some sentences require editing:

Introduction:

1. Critically short telomeres, induced by breeding of multiple generations ageing? mice lacking the catalytic subunit of telomerase Terc, leads to...
2. As such, the mechanisms that drive senescence in postmitotic cells and the contribution that?

postmitotic cell senescence (PoMiCS) in tissue degeneration, including the heart....

3. we also found an age-dependent increase in TAF (but not non-TAF) was observed? in other post-mitotic cells.....

4. Telomeres are repetitive sequences of DNA, associated? a protein complex known as shelterin....

Stress-induced telomere length
Senescence-like phenotype in CM

Referee #3:

This is a manuscript outlining the role of telomere damage in the regulation of cardiomyocyte senescence, hypertrophy and regeneration. The overall premise of the study is that senescent cardiomyocytes are responsible for the overall senescence of the heart through SASPs. The authors show that telomere dysfunction increases with age in cardiomyocytes. The manuscript suffers from several major flaws including lack of novelty, and lack of technical rigor in particular with regards to major claims of cardiomyocyte mitosis and the degree of senescence of cardiomyocytes.

Major concerns:

1) The observation that telomere dysfunction occurs in cardiomyocytes is not new. Ignacio Flores showed that this starts in neonatal cardiomyocytes and that it mediates cessation of cell division in cardiomyocytes. (Aix et al. *Journal of Cell Biology*. 2016. Telomere shortening limits the capacity of the heart to regenerate)

2) The data in figure 2 regarding the effect of radiation, while interesting, is irrelevant to the topic of this manuscript. I understand that the authors are trying to say that severe DNA damage in cardiomyocytes persists, but with 10Gy, this is completely irrelevant to the topic spontaneous DNA damage.

3) The data in figure 3 don't make sense. Only 30% of cardiomyocytes are p21 positive at 3 months of age? Previous reports almost two decades ago showed that over 75% of cardiomyocytes are p21 positive early on in the early postnatal period by postnatal day 6 (Horky et al. *Phys. Res.* 1998 Induction of Cell-Cycle Inhibitor p21 in Rat Ventricular Myocytes during Early Postnatal Transition from Hyperplasia to Hypertrophy). This is also consistent with previous reports such as Puente et al (*Cell*, 2014. The oxygen-rich postnatal environment induces cardiomyocyte cell-cycle arrest through DNA damage response), which showed the induction of DNA damage response proteins such as ATM in the early postnatal heart. There are numerous literature reports suggesting that p16 and p21 and other DNA damage response regulators are expressed in the early postnatal heart. So the premise that this is exclusively an aging heart phenotype is not supported by the literature.

4) Figure 7 and 8 represent the weakest set of data for several reasons. 1) The authors claim that by 30 months, close to 80% of cardiomyocytes are senescent by expression of p21. How is it possible that elimination of these senescent cardiomyocytes does not cause immediate heart failure? The only explanation is that either the p21 expressing myocytes are not senescent, or that the genetic and pharmacological manipulation does not in fact eliminate even a fraction of senescent cardiomyocytes. 2) The studies performed to quantify mitosis of myocytes are very poor. The arbitrary drawing of myocyte borders is unacceptable and WGA should be used in conjunction with a cardiac marker to detect myocytes and their borders. Also, the aurora b kinase shown in the figure is expressed between two nuclei which is a marker of karyokinesis, not cytokinesis. Also, ki67 is not a reliable marker of proliferation. Finally, for this substantial claim to be proven, additional studies such as using the MADM mouse (similar to recent *Cell* paper by the Srivastav group) as well as cardiomyocyte count and an injury model followed by regeneration have to be employed.

We would like to thank the reviewers for their time and insightful comments which we believe have helped improve the quality of our study.

We have made every effort to address the reviewer's comments; both through the inclusion of new experimental data, and the inclusion of additional detail and discussion.

Please see our point-by-point response to referees below.

Referee #1:

This is an interesting manuscript describing the accumulation of persistent DNA damage at telomeres in cardiomyocytes and improved health upon removal of senescent cells.

I have relatively few suggestions to improve this manuscript:

- Telomere clustering has not been fully ruled out in my opinion. Measuring the number of telomeric signals per cell may give an indication that indeed the techniques employed really allow the detection of all 92 individual telomeres expected in a nucleus thus excluding clustering that could reduce their apparent number and artefactually increase their length.

We thank the reviewer for the important point. All Super-resolution imaging was done in 3µm sections in interphase nuclei so we cannot capture an entire CM nucleus- therefore we can only detect a lower number of telomeres than expected.

We should like to highlight that in the conditions we acquired the STED images we gained 3x resolution on XY and nearly 2.3x resolution in Z compared to standard confocal. Resolution was further increased with a dedicated STED deconvolution algorithm. However, we agree with the reviewer that even STED has resolution limitations that cannot be completely overcome however, the detection power of STED is still, in our view, the best method we have to identify individual telomeres on whole nuclei and discern telomere clusters.

We have expanded the discussion to highlight these particular methodological pitfalls and replaced some of text to make sure that we do not over-interpret our results.

We include the comparison in Fig EV1E of the average number of telomere signals we observe by STED microscopy vs confocal (showing increased number of telomere signals by STED).

We have also avoided statements such as "STED resolved clustered telomeres" and replaced it with "STED improved the resolution of clustered telomeres".

- Fig S1b needs to be complemented by a quantification of 53bp1 total number of foci as in 1d

We thank the reviewer- we agree this is an important control to include. It is now part of Fig EV1B. In summary, we find no differences in the total number of 53BP1 foci between 4 and 24m old mice (similar to our results with gH2A.X).

- ChIP in 1f need a negative control/input normalizer: Alu PCR?

We determined the differences in the DNA content of the bound and input fractions. Bound to input (B/I) ratios were determined for each amplicon by taking a fixed aliquot of the DNA extracted from the input and bound samples. These were amplified together with a defined amount of genomic DNA on the same plate, using the latter to construct the standard curve. B/I values for any one amplicon are thus in the correct quantitative ratio to B/I values for other amplicons measured for the same ChIP, because this procedure compensates for differences in the PCR efficiencies of different probe/primer combinations and for result representation. With each of the samples, we also included an isotype matched, irrelevant antibody control which was subtracted from the result before plotting the results. Using this combined approach eliminates the need to include a negative gene control.

We have added this information to the detailed methods to clarify this in the manuscript.

- Irreparability of telomeres was demonstrated at the molecular level by Fumigalli et al and it would be fair to quote this along Hewitt et al wherever this report is duly quoted

The reviewer is correct and we apologise for this. We have made sure we cited the Fumagalli et al. 2012 paper together with Hewitt et al. 2012. This was a mistake on our part and has been rectified.

- Experiments with TRF1-Fok1 are interesting but lack an adequate control: a nuclease that induces a similar number of DSB in not telomeric regions. That is essential to support the claims made.

We thank the reviewer for this excellent point. We have transfected an inducible endonuclease I-PpoI which upon short treatment with tamoxifen induced a very similar number of DDR foci (at non telomeric regions) in neonatal cardiomyocytes to the number of TAF which we found upon expression of TRF1-FOKI. Upon removal of tamoxifen we found that majority of DNA damage foci became repaired (Fig EV2I). In these cells we did not observed any induction of senescent markers such as SA-b-Gal or increased cell size.

In contrast, when we induced telomeric DNA damage (using an inducible TRF1-FOKI), damage was unrepaired and cells acquired senescent markers (Fig EV2G, H).

- I would recommend complementing the results obtained with conditioned medium from old CM by testing its impact on DDR and senescence markers: are they induced?

We thank the reviewer for this excellent suggestion. As suggested, we have treated mouse adult fibroblasts with conditioned medium isolated from old CMs. We found a significant increase in SA-b-Gal activity and a decrease in EdU positive cells. We observed a tendency for an increase in the number of 53BP1 foci, however, not statistically significant. We also confirmed that treatment with conditioned media from old CMs also did not significantly impact on DDR foci in cardiac fibroblasts (but reduced EdU incorporation). These data are now included in Appendix Figure S5 E-G.

- It may be appropriate to quote this recent publication
<https://www.ncbi.nlm.nih.gov/pubmed/30150400>

We thank the reviewer for the excellent suggestion – we have cited the paper.

Referee #2:

Reductions in telomere length have been implicated in cellular senescence. The authors address the question how ageing affects cellular senescence in postmitotic cardiac myocytes, which have a very low proliferative activity and thus telomere shortening is unlikely to be a trigger in these cells. An impressive amount of experimental data using ageing mice and an additional 6 KO and transgenic lines plus primary cultures, cardiac cell lines and a total of 8 primary panels and 12 supplements, all packed with data lead to very convincing conclusion that mitochondria derived reactive oxygen species probably trigger DNA damage leading to the binding of proteins of the DNA damage response such as γ H2A.X, 53BP1 to the telomere triggering a senescent program the animals demonstrated an age-dependent increase in expression of CDK inhibitors p16INK4a, p21CIP, and p15 INK4a in cardiac myocytes. Senescence in proliferating cells is also associated with a specific secretory pathway, which however seems to be absent in cardiac myocytes. Nonetheless some cytokines such as endothelin 3, TGF β 2 and GDF15 were found to be upregulated in old hearts. Using a number of genetic and experimental approaches the authors show that the trigger of age - dependent cardiac myocyte senescence appears to be enhanced mitochondrial ROS activity. Senescent hearts show an increase in cardiac hypertrophy, wall rigidity and fibrosis while ejection fraction is preserved. Genetic and pharmacological approaches were developed to remove specifically senescent cells from the ageing heart, which normalize cardiac fibrosis and cardiac hypertrophy. Some of the data suggest that the clearance of senescent myocytes may trigger activation of resident stem cells to replenish the cardiac myocyte pool with young myocytes as total cell number and the size of the ventricular wall remain unchanged.

The experiments have been executed to a very high standard and apart from a quantification of the reduction in fibrosis in the genetic approach to remove senescent cells (INK-ATTAC), I have no specific suggestions for additional experiments.

We really appreciate that the reviewer finds our results important and well executed. We have quantified reduced fibrosis in INK-ATTAC mice – it is included in Figure 7G.

Minor

some sentences require editing:

Introduction:

1. Critically short telomeres, induced by breeding of multiple generations ageing? mice lacking the catalytic subunit of telomerase Terc, leads to...
2. As such, the mechanisms that drive senescence in postmitotic cells and the contribution that? postmitotic cell senescence (PoMiCS) in tissue degeneration, including the heart....
3. we also found an age-dependent increase in TAF (but not non-TAF) was observed? in other post-mitotic cells.....
4. Telomeres are repetitive sequences of DNA, associated? a protein complex known as shelterin....

Stress-induced telomere length
Senescence-like phenotype in CM

We thank the reviewer for spotting these mistakes. All have been corrected in the revised version.

Referee #3:

This is a manuscript outlining the role of telomere damage in the regulation of cardiomyocyte senescence, hypertrophy and regeneration. The overall premise of the study is that senescent cardiomyocytes are responsible for the overall senescence of the heart through SASPs. The authors show that telomere dysfunction increases with age in cardiomyocytes. The manuscript suffers from several major flaws including lack of novelty, and lack of technical rigor in particular with regards to major claims of cardiomyocyte mitosis and the degree of senescence of cardiomyocytes.

Major concerns:

- 1) The observation that telomere dysfunction occurs in cardiomyocytes is not new. Ignacio Flores showed that this starts in neonatal cardiomyocytes and that it mediates cessation of cell division in cardiomyocytes. (Aix et al. Journal of Cell Biology. 2016. Telomere shortening limits the capacity of the heart to regenerate)

We thank the reviewer for highlighting this paper. The cited paper shows that after birth cardiomyocytes lose telomerase activity and telomere length which contributes to a certain degree of telomere dysfunction (authors measured co-localisation of gH2A.X and telomeres at P8 and found around 5%). This is not inconsistent with our own analysis of very young animals and in our view does not influence the novel observation that with age the % of cardiomyocytes containing TAF increases.

- 2) The data in figure 2 regarding the effect of radiation, while interesting, is irrelevant to the topic of this manuscript. I understand that the authors are trying to say that severe DNA damage in cardiomyocytes persists, but with 10Gy, this is completely irrelevant to the topic spontaneous DNA damage.

We understand the reasoning of the reviewer, but we believe these data are an important contributor to our conclusions. These data was added with the purpose of highlighting that telomeres when damaged are more difficult to repair than non-telomeric regions. While similar experiments have been performed in other cell types such as fibroblasts (Fumagalli et al. 2012; Hewitt et al. 2012), they have not been performed in cells of a cardiac origin. The use of X-ray irradiation is commonly used in the senescence field- as a way to induce senescence (which requires relatively high doses in order to generate a homogeneous population of senescent cells). In fact, given the fact that telomeres occupy a very small fraction of the entire genome, we require the use of high doses of irradiation or oxidative stress agents to generate TAF randomly in vitro and in a short period of time.

3) The data in figure 3 don't make sense. Only 30% of cardiomyocytes are p21 positive at 3 months of age? Previous reports almost two decades ago showed that over 75% of cardiomyocytes are p21 positive early on in the early postnatal period by postnatal day 6 (Horky et al. Phys. Res. 1998 Induction of Cell-Cycle Inhibitor p21 in Rat Ventricular Myocytes during Early Postnatal Transition from Hyperplasia to Hypertrophy). This is also consistent with previous reports such as Puente et al (Cell, 2014. The oxygen-rich postnatal environment induces cardiomyocyte cell-cycle arrest through DNA damage response), which showed the induction of DNA damage response proteins such as ATM in the early postnatal heart. There are numerous literature reports suggesting that p16 and p21 and other DNA damage response regulators are expressed in the early postnatal heart. So the premise that this is exclusively an aging heart phenotype is not supported by the literature.

The reviewer is correct that an induction of the DDR response in cardiomyocytes has been demonstrated early in the postnatal period at day 5-6 after birth (Puente et al, Cell, 2014). This DDR is responsible for the arrest of cell proliferation of neonatal cardiomyocytes and leads to the induction of cell cycle inhibitors such as p21.

However, further studies clearly demonstrated that this developmental induction of p21 is a transient process and that its expression returns to baseline levels in adults (see publications by Aix, JCB 2016; Tane, BBRC 2014).

This is consistent with our observations that p21 expression in young adult hearts is relatively low and increases with ageing. It is possible that a certain degree of p21 is expressed in most cardiomyocytes and this would indeed be consistent with our own data which shows that majority of cardiomyocytes contain 3-4 γ H2A.X and 53BP1 foci (which could activate the p53/p21 pathway). However, we have confirmed our data using both IHC as well as RT-PCR (for p21 mRNA) in isolated cardiomyocytes which show clearly that there is an age-dependent increase in p21. Our IHC analysis of p21 levels was done in a blinded fashion and reproduced independently by different observers. Additionally, we have recently reported an increase in p21 expression by a different method (western blotting) in ventricular cardiomyocytes isolated from old mice from a completely different cohort (Manzella et al. Aging Cell 2018). Other groups have also reported increased expression of p21 with age in heart (Baker et al. 2016).

4) Figure 7 and 8 represent the weakest set of data for several reasons. 1) The authors claim that by 30 months, close to 80% of cardiomyocytes are senescent by expression of p21. How is it possible that elimination of these senescent cardiomyocytes does not cause immediate heart failure? The only explanation is that either the p21 expressing myocytes are not senescent, or that the genetic and pharmacological manipulation does not in fact eliminate even a fraction of senescent cardiomyocytes.

While increased P21 expression is associated with senescence, p21 does not unambiguously indicate a senescent state. It is well established that p21 can be expressed in a transient manner in cells which are temporarily arrested. In fact, there is a consensus in the field that there is no universal senescent marker- thus a combination of various markers should be conducted. For that reason, we have analysed TAF, p21, SADS, p16, p15 in cardiomyocytes from aged animals.

We do not claim that all cardiomyocytes containing TAF or p21 are senescent. In fact, the number of TAF required to induce a senescent phenotype is under debate with some studies indicating that merely 1 dysfunctional telomere (di Leonardo et al. 1994) is sufficient to arrest cell growth while more recent data indicating that at least 5 dysfunctional telomeres are necessary (Kaul et al. 2012). This is likely to vary between different cell types. Furthermore, recent publications have suggested that the % of cells positive for ≥ 3 TAF may be a better indication of senescence, at least in some tissues (see publications by Ogrodnik et al. 2017, Jurk et al. 2014).

In fact, when we analysed cardiomyocytes in old animals containing at least 3 TAF we found around 20% and this value was reduced to 5% following pharmacogenetics and pharmacological clearance of senescent cells (Figure 7E and Figure 8B). Presently, probably the marker of senescence which is considered the most robust is p16^{ink4a} (which has been associated with the irreversibility of the senescence-arrest). When conducting RNA-ISH, we could only observe that around 20% of cardiomyocytes were positive for p16 mRNA and this level was reduced to 5% following treatment with AP (this was confirmed by conducting RNA-ISH against eGFP which is expressed as part of the INK-ATTAC transgene). Thus, it is likely that only a relatively small % of cells is being lost.

Additionally, a multitude of other studies have reported clearance of senescent cells using genetic and pharmacological approaches without any adverse changes in cardiac function (see for eg. Baker, D. J. et al 2016, Demaria, M. et al 2011, Zhu, Y. et al 2015).

Finally, we are aware of a complementary study from another lab who independently conducted similar analysis using the INK-ATTAC model as well as another senolytic drug cocktail (dasatinib and quercetin). They reported reduced p16 in the heart without major changes in heart function (BioRxiv doi: <https://doi.org/10.1101/397216>) and similarly to us- observed a reduction in hypertrophy and fibrosis.

2) The studies performed to quantify mitosis of myocytes are very poor. The arbitrary drawing of myocyte borders is unacceptable and WGA should be used in conjunction with a cardiac marker to detect myocytes and their borders.

We apologise to the reviewer for lack of clarity in the methodology we used. We have in fact used WGA in conjunction with cardiac markers to detect myocytes and their borders as shown in Figure 8H and movies showing images throughout the z-stack series are now included as supplementary data.

To measure cardiomyocyte proliferation we followed protocols similar to those used in several different publications: (see e.g. Puente *et al* Cell 2014, Xin et al PNAS 2013, Senyo *et al* Nature 2013, Vujic et al Nature Comm 2018, Zacchigna, et al Nature Comm. 2018).

Also, the aurora b kinase shown in the figure is expressed between two nuclei which is a marker of karyokinesis, not cytokinesis. Also, ki67 is not a reliable marker of proliferation. Finally, for this substantial claim to be proven, additional studies such as using the MADM mouse (similar to recent Cell paper by the Srivastav group) as well as cardiomyocyte count and an injury model followed by regeneration have to be employed.

The reviewer is correct in stating that Aurora b is expressed during binucleation as well as cytokinesis. We would however like to point out the recently published study that provides evidence that the location of Aurora b differs during these different processes (Circulation Research. 2018;123:1039–1052 and editorial Circulation Research. 2018;123:1012–1014). This work suggests that Aurora b expression between two nuclei is indicative of cytokinesis (as we observed), while an asymmetrical location of Aurora b is indicative of binucleation.

While, we agree that Ki67 alone is not a reliable marker of proliferation, we would however suggest that the Ki67 data taken together with the appearance of smaller cardiomyocytes, the expression of Aurora b in the mid-body and the significant increase in mononuclear cardiomyocytes which have incorporated Edu is suggestive of increased cardiomyocyte generation (which we interpret as a possible compensation for the cell loss following senescent cell elimination).

Importantly, another group has independently conducted similar experiments (available in BioRxiv doi: <https://doi.org/10.1101/397216>) and showed that clearance of senescent cells (genetically or using different senolytic drugs) was accompanied by increased EdU incorporation and expression of proliferation markers in CMs. This increases our confidence in the reproducibility of our results.

With regards to the use of an injury model suggested by the reviewer we fail to see how these experiments would benefit our study. To clarify, we are not suggesting that the removal of senescent cells increases the regenerative potential of the heart. As stated, these experiments were merely an attempt to explain why despite an apparent loss of senescent cardiomyocytes we did not observe any loss of heart function.

Similarly, while we agree with the reviewer that the MADM mouse is an excellent model to detect CM proliferation, it would not be realistic in the time frame awarded for the revisions to conduct the suggested experiments. These would require complex breeding and ageing of mice and we estimate would take at least 3 years.

As suggested by the reviewer, we have highlighted possible methodological pitfalls in our experimental approach in the discussion and suggest compensatory proliferation following clearance of senescent cells as one possible interpretation of our data.

2nd Editorial Decision

14th Dec 2018

Thank you for submitting the revised version of your manuscript. Your revised study has now been re-evaluated by the three original referees and we have received comments from two of them, which I enclose below. Please note that while referee #3 was not able to look back into the work at this time, we have asked the other two referees to consider your response to his/her concerns as well and have in addition editorially assessed this matter.

As you will see the referees find that their concerns have been sufficiently addressed and they are now broadly favour of publication.

Thus, we are pleased to inform you that your manuscript has been accepted in principle for publication in The EMBO Journal, pending some minor issues regarding material and methods, formatting and data representation, as outlined below, which need to be adjusted at re-submission.

REFEREE REPORTS:

Referee #1:

I am happy with the revised manuscript. It can be published as is

Referee #2:

Reductions in telomere length have been implicated in cellular senescence. The authors address the question how ageing affects cellular senescence in postmitotic cardiac myocytes, which have a very low proliferative activity and thus telomere shortening is unlikely to be a trigger in these cells. An impressive amount of experimental data using ageing mice and an additional 6 KO and transgenic lines plus primary cultures, cardiac cell lines and a total of 8 primary panels and 12 supplements, all packed with data, lead to very convincing conclusion that mitochondria-derived reactive oxygen species trigger a senescence program in cardiac myocytes. Senescent hearts show an increase in cardiac hypertrophy, wall rigidity and fibrosis while ejection fraction is preserved. Genetic and pharmacological approaches were developed to remove senescent cells from the aging heart, which normalize cardiac fibrosis and cardiac hypertrophy.

The revised version has addressed satisfactorily my concerns toward the original manuscript.

2nd Revision - authors' response

27th Nov 2018

All requested editorial changes were made.

Gavin Richardson

EMBOJ

Manuscript Number: EMBOJ-2018-100492